# Improving equilibrium propagation without weight symmetry through Jacobian homeostasis

**Axel Laborieux**[1], **Friedemann Zenke**[1,2]
[1]Friedrich Miescher Institute for Biomedical Research, Basel, Switzerland
[2]Faculty of Science, University of Basel, Switzerland
`{firstname.lastname}@fmi.ch`

## Abstract

Equilibrium propagation (EP) is a compelling alternative to the backpropagation of error algorithm (BP) for computing gradients of neural networks on biological or analog neuromorphic substrates. Still, the algorithm requires weight symmetry and infinitesimal equilibrium perturbations, i.e., nudges, to yield unbiased gradient estimates. Both requirements are challenging to implement in physical systems. Yet, whether and how weight asymmetry contributes to bias is unknown because, in practice, its contribution may be masked by a finite nudge. To address this question, we study generalized EP, which can be formulated without weight symmetry, and analytically isolate the two sources of bias. For complex-differentiable non-symmetric networks, we show that bias due to finite nudge can be avoided by estimating exact derivatives via a Cauchy integral. In contrast, weight asymmetry induces residual bias through poor alignment of EP's neuronal error vectors compared to BP resulting in low task performance. To mitigate the latter issue, we present a new homeostatic objective that directly penalizes functional asymmetries of the Jacobian at the network's fixed point. This homeostatic objective dramatically improves the network's ability to solve complex tasks such as ImageNet $32{\times}32$. Our results lay the theoretical groundwork for studying and mitigating the adverse effects of imperfections of physical networks on learning algorithms that rely on the substrate's relaxation dynamics.

## 1 Introduction

Virtually all state-of-the-art artificial intelligence (AI) models are trained using the backpropagation of error algorithm (BP) (Rumelhart et al., 1986). In BP, neural activations are evaluated during the forward pass, and gradients are obtained through a corresponding backward pass. While BP can be implemented efficiently on digital hardware which *simulates* neural networks, it is less suitable for physical neural networks such as brains or neuromorphic substrates, which operate at lower energy costs. This limitation is mainly due to two technical requirements of the backward pass in BP. First, the backward pass is linear while leaving neuronal activity unaffected (Lillicrap et al., 2020; Whittington & Bogacz, 2019). Second, back-propagation requires the transpose of connectivity matrices, often referred to as "weight symmetry" (Grossberg, 1987). For these reasons, many alternative learning algorithms that can run on dedicated neuromorphic hardware have been designed (Boybat et al., 2018; Thakur et al., 2018; Indiveri et al., 2011; Schuman et al., 2017; Frenkel & Indiveri, 2022; Yi et al., 2023).

One such alternative learning algorithm is equilibrium propagation (EP)(Scellier & Bengio, 2017), which can provably estimate gradients using *only* the network's own dynamics. Classic EP requires an energy-based model (EBM), such as a differentiable Hopfield network (Hopfield, 1984), which relaxes to an equilibrium for any given input. EP computes gradients by comparing neuronal activity at the free equilibrium to the activity at a second equilibrium that is "nudged" toward a desired target output. Preliminary hardware realization dedicated to EP suggests that it could reduce the energy cost of AI training by four orders of magnitude (Yi et al., 2023). Unlike BP in feed-forward networks, EP uses the *same* network to propagate feed-forward and feed-back activity, thereby dispensing with

the need to linearly propagate error vectors. In particular, EP is a promising candidate for learning through neuronal oscillations (Baldi & Pineda, 1991; Laborieux & Zenke, 2022; Anisetti et al., 2022; Delacour et al., 2021), which is why we focus on EP in this work.

Nevertheless, EP requires weight symmetry and vanishing nudge for unbiased gradient estimates, which limits its potential for neuromorphics compared to other better performing BP alternatives (Ren et al., 2023; Journé et al., 2023; Payeur et al., 2021; Greedy et al., 2022; Høier et al., 2023). While Laborieux & Zenke (2022) showed that an oscillation-based extension of EP called holomorphic EP (hEP) removes the bias by integrating oscillations of neuronal activity, this approach has only been demonstrated using symmetric weights. Although a generalization of EP to non-symmetric dynamical systems exists, the approach has only been demonstrated on simple tasks like MNIST (Scellier et al., 2018; Ernoult et al., 2020; Kohan et al., 2018), while it fails to train on CIFAR-10 (Laborieux et al., 2021). Previous work investigated the effect of non-symmetric feedback on training with BP (Lillicrap et al., 2016; Nøkland, 2016; Launay et al., 2020; Clark et al., 2021; Refinetti et al., 2021) and approximate BP (Hinton, 2022; Dellaferrera & Kreiman, 2022; Löwe et al., 2019), but little attention has been given to effect of weight symmetry for EP. In this article we fill this gap. Overall, our main contributions are:

- We provide a comprehensive analysis of the individual sources of bias in the gradient estimate from weight asymmetry and finite-size nudge in generalized EP.
- An extension of hEP (Laborieux & Zenke, 2022) to non symmetric complex-differentiable dynamical systems, that can be estimated through continuous oscillations.
- We propose a new homeostatic loss that reduces the asymmetry of the Jacobian at the free equilibrium point without enforcing perfect weight symmetry.
- An empirical demonstration that hEP with homeostatic loss scales to ImageNet $32\times32$, with a small performance gap compared to the symmetric case.

## 2 BACKGROUND

### 2.1 GRADIENTS IN CONVERGING DYNAMICAL SYSTEMS

Let us first recall how gradients are computed in converging dynamical systems (Baldi, 1995). Let $F$ be a differentiable dynamical system with state variable $\boldsymbol{u}$ and parameters $\boldsymbol{\theta}$. A subset of units receive static input currents $\boldsymbol{x}$. The dynamics of the system are governed by:

$$\frac{\mathrm{d}\boldsymbol{u}}{\mathrm{d}t} = F(\boldsymbol{\theta}, \boldsymbol{u}, \boldsymbol{x}). \tag{1}$$

We further assume that the dynamics of Eq. (1) converge to a fixed point of the state $\boldsymbol{u}$. This phase of the dynamics is defined as the "free phase", because there are no contributions from any target $\boldsymbol{y}$ associated to $\boldsymbol{x}$. While stable fixed points provably exists for EBMs (Hopfield, 1984), there is no guarantee to converge for arbitrary dynamical systems. However, a fixed point exists if $F$ is a contraction map (Scarselli et al., 2008; Liao et al., 2018). In the following, we just assume the existence of a fixed point and denote it by $\boldsymbol{u}_0^*$:

$$0 = F(\boldsymbol{\theta}, \boldsymbol{u}_0^*, \boldsymbol{x}). \tag{2}$$

In $\boldsymbol{u}_0^*$, the superscript $*$ indicates convergence to a fixed point, and the subscript 0 stands for the "free" phase. We further assume that $\boldsymbol{u}_0^*$ is an implicit function of $\boldsymbol{\theta}$, such that its derivative exist by means of the implicit function theorem (IFT). Given an objective function $\mathcal{L}$ that measures the proximity of a subset of $\boldsymbol{u}^*$ (output units) to a target $\boldsymbol{y}$, the gradient of $\mathcal{L}$ with respect to $\boldsymbol{\theta}$ is obtained by the chain rule through the fixed point $\frac{\mathrm{d}\mathcal{L}}{\mathrm{d}\boldsymbol{\theta}} = \frac{\mathrm{d}\mathcal{L}}{\mathrm{d}\boldsymbol{u}_0^*}\frac{\mathrm{d}\boldsymbol{u}_0^*}{\mathrm{d}\boldsymbol{\theta}}$, where $\frac{\mathrm{d}\mathcal{L}}{\mathrm{d}\boldsymbol{u}_0^*}$ is the error at the output units, and $\frac{\mathrm{d}\boldsymbol{u}_0^*}{\mathrm{d}\boldsymbol{\theta}}$ is obtained by differentiating Eq. (2) with respect to $\boldsymbol{\theta}$:

$$0 = \frac{\partial F}{\partial \boldsymbol{\theta}}(\boldsymbol{u}_0^*) + J_F(\boldsymbol{u}_0^*) \cdot \frac{\mathrm{d}\boldsymbol{u}_0^*}{\mathrm{d}\boldsymbol{\theta}},$$
$$\frac{\mathrm{d}\boldsymbol{u}_0^*}{\mathrm{d}\boldsymbol{\theta}} = -J_F(\boldsymbol{u}_0^*)^{-1} \cdot \frac{\partial F}{\partial \boldsymbol{\theta}}(\boldsymbol{u}_0^*). \tag{3}$$

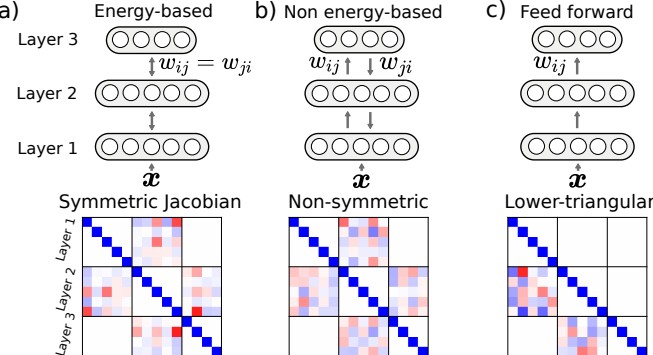

Figure 1: Different neural network architectures and their Jacobian after the free phase. **a)** A continuous layered Hopfield network. The existence of an energy function enforces Jacobian symmetry with reciprocal and tied connectivity $w_{ij} = w_{ji}$. **b)** A layered network with reciprocal connectivity but independent forward and backward connections, which make the Jacobian not symmetric. **c)** A discrete feed forward network without feedback connections. The Jacobian is lower triangular, allowing the explicit back propagation of error layer by layer.

Where $J_F(\boldsymbol{u}_0^*) := \frac{\partial F}{\partial \boldsymbol{u}}(\boldsymbol{\theta}, \boldsymbol{u}_0^*, \boldsymbol{x})$ is the Jacobian of the network at the fixed point (Fig. 1). Replacing $\frac{\mathrm{d}\boldsymbol{u}_0^*}{\mathrm{d}\boldsymbol{\theta}}$ in the expression of the gradient $\frac{\mathrm{d}\mathcal{L}}{\mathrm{d}\boldsymbol{\theta}}$ and transposing the whole expression yields (Baldi, 1995):

$$\frac{\mathrm{d}\mathcal{L}}{\mathrm{d}\boldsymbol{\theta}}^\top = \underbrace{\frac{\partial F}{\partial \boldsymbol{\theta}}(\boldsymbol{u}_0^*)^\top}_{\text{"pre-synaptic"}} \cdot \underbrace{\left(-J_F(\boldsymbol{u}_0^*)^{-\top} \cdot \frac{\mathrm{d}\mathcal{L}}{\mathrm{d}\boldsymbol{u}_0^*}^\top\right)}_{\text{"post-synaptic" } \boldsymbol{\delta}}. \tag{4}$$

We refer to the part of the expression in parenthesis as the neuronal error vector $\boldsymbol{\delta}$. In the widely used reverse-mode automatic differentiation (AD), a linear system involving the transpose of the Jacobian is solved to compute $\boldsymbol{\delta}$ from the output error $\frac{\mathrm{d}\mathcal{L}}{\mathrm{d}\boldsymbol{u}_0^*}$, and finally multiply with $\frac{\partial F}{\partial \boldsymbol{\theta}}(\boldsymbol{u}_0^*)$, which is a sparse matrix involving pre-synaptic variables. In feed forward networks, the lower-triangular structure of the Jacobian allows computing $\boldsymbol{\delta}$ recursively layer by layer from the output (Fig. 1c) as exploited in BP (Rumelhart et al., 1986). In converging dynamical systems, recurrent backpropagation (RBP) and variants (Almeida, 1990; Pineda, 1987; Liao et al., 2018) obtain $\boldsymbol{\delta}$ as the fixed point of an auxiliary linear dynamical system using $J_F(\boldsymbol{u}_0^*)^\top$. In deep equilibrium models (Bai et al., 2019) $\boldsymbol{\delta}$ is found with a root-finding algorithm. In the next section, we review how $\boldsymbol{\delta}$ is obtained in EP.

## 2.2 NEURONAL ERRORS IN EQUILIBRIUM PROPAGATION

The EP (Scellier & Bengio, 2017) gradient formula for EBMs can be derived without explicit use of the energy function formalism (Scellier et al., 2018), by noticing a similar pattern between the derivative of $\boldsymbol{u}_0^*$ with respect to parameters $\boldsymbol{\theta}$ (Eq. (3)) and $\boldsymbol{\delta}$ in Eq. (4). While $\frac{\mathrm{d}\boldsymbol{u}_0^*}{\mathrm{d}\boldsymbol{\theta}}$ is obtained by inversion of $J_F(\boldsymbol{u}_0^*)$, $\boldsymbol{\delta}$ is obtained by inversion of $J_F(\boldsymbol{u}_0^*)^\top$. However, in EBMs for which there exists a scalar energy function $E$ such that $F = -\frac{\partial E}{\partial \boldsymbol{u}}$, we have precisely $J_F(\boldsymbol{u}_0^*) = J_F(\boldsymbol{u}_0^*)^\top$ (Fig. 1a) due to the Schwarz theorem on symmetric second derivatives. The remaining difference between Eqs. (3) and (4) is the quantity to which the inverted Jacobian is applied: $\frac{\partial F}{\partial \boldsymbol{\theta}}(\boldsymbol{u}_0^*)$ in Eq. (3) and $\frac{\mathrm{d}\mathcal{L}}{\mathrm{d}\boldsymbol{u}_0^*}^\top$ in Eq. (4). The solution is to define a parameter $\beta$ such that these quantities are equal:

**Definition 1** (Nudge parameter $\beta$). *Let $\beta$ be a scalar parameter that is equal to zero during the free phase, justifying the notation $\boldsymbol{u}_0^*$, and such that $\frac{\partial F}{\partial \beta}(\boldsymbol{u}_0^*) = \frac{\mathrm{d}\mathcal{L}}{\mathrm{d}\boldsymbol{u}_0^*}^\top$. When $\beta \neq 0$, the target $\boldsymbol{y}$ contributes to the dynamics through $\frac{\mathrm{d}\mathcal{L}}{\mathrm{d}\boldsymbol{u}_0^*}$. The term "teaching amplitude" refers to $|\beta|$.*

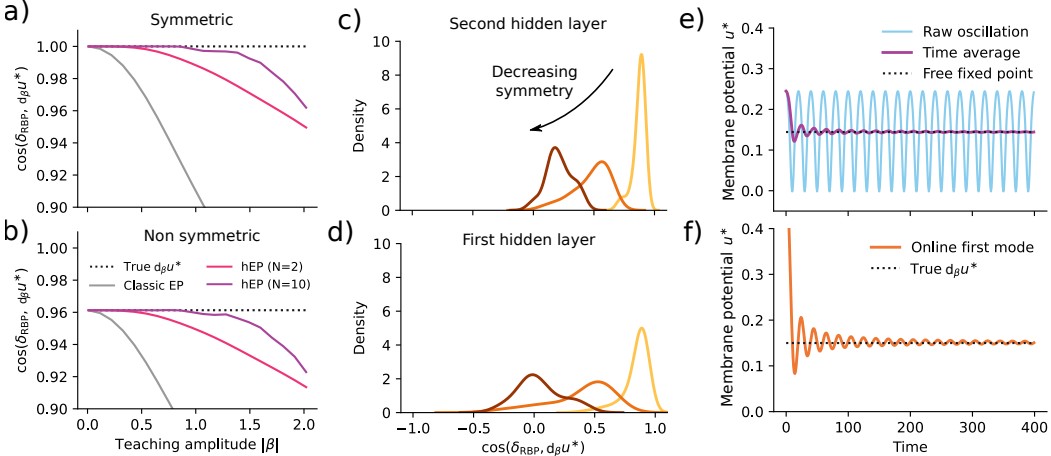

Figure 2: Separating the sources of bias due to finite nudge and Jacobian asymmetry. **a,b)** Cosine similarity between the neuronal error vector $\delta$ computed by RBP and classic EP (grey) (Scellier & Bengio, 2017; Scellier et al., 2018), holomorphic EP (purple and pink) for different number of points to estimate Eq. (7), in function of the teaching amplitude $|\beta|$, for symmetric (**a)**) and non symmetric (**b)**) equilibrium Jacobian. **c,d)** Kernel density estimate of the cosine distribution between both neuronal error vectors in a two-hidden layer MLP. Alignment worsens with increasing asymmetry and depth. **e)** Continuous-time estimate of the free fixed point through oscillations induced by the teaching signal $\beta(t)$. **f)** Continuous-time estimate of the neuronal error vector of generalized hEP.

By applying the same derivation as Eq. (3) for the new parameter $\beta$, we have:

$$\frac{\mathrm{d}\boldsymbol{u}^*}{\mathrm{d}\beta}\bigg|_{\beta=0} = -J_F(\boldsymbol{u}_0^*)^{-1} \cdot \frac{\partial F}{\partial \beta}(\boldsymbol{u}_0^*) = -\underbrace{J_F(\boldsymbol{u}_0^*)^{-\top}}_{\text{for EBMs}} \cdot \underbrace{\frac{\mathrm{d}\mathcal{L}}{\mathrm{d}\boldsymbol{u}_0^*}^{\top}}_{\text{by Def. 1}} = \boldsymbol{\delta}. \tag{5}$$

Therefore, the EP neuronal error vector is the derivative of the fixed point $\boldsymbol{u}_0^*$ with respect to $\beta$ in $\beta = 0$, which we denote by $\mathrm{d}_\beta \boldsymbol{u}^*$ for short. Note that the derivation assumes that $\boldsymbol{u}_0^*$ is an implicit function of $\beta$. In practice, $\mathrm{d}_\beta \boldsymbol{u}^*$ is approximated by finite differences after letting the network settle to a second equilibrium $\mathrm{d}_\beta \boldsymbol{u}^* \approx (\boldsymbol{u}_\beta^* - \boldsymbol{u}_0^*)/\beta$, which contains a bias due to the finite $\beta \neq 0$ used in the second phase. The EP gradient estimate can then be written without using the energy function by replacing $\boldsymbol{\delta}$ by $\mathrm{d}_\beta \boldsymbol{u}^*$ in Eq. (4):

$$\tilde{\nabla}_{\boldsymbol{\theta}} := \frac{\partial F}{\partial \boldsymbol{\theta}}(\boldsymbol{u}_0^*)^{\top} \cdot \mathrm{d}_\beta \boldsymbol{u}^*. \tag{6}$$

Importantly, the quantities $\mathrm{d}_\beta \boldsymbol{u}^*$ and $\tilde{\nabla}_{\boldsymbol{\theta}}$ are defined for *any* differentiable dynamical system (Scellier et al., 2018), not just for EBMs. However, $\mathrm{d}_\beta \boldsymbol{u}^*$ coincides with $\boldsymbol{\delta}$, and $\tilde{\nabla}_{\boldsymbol{\theta}}$ with $\frac{\mathrm{d}\mathcal{L}}{\mathrm{d}\boldsymbol{\theta}}$ (Eq. (4)), only for EBMs as their Jacobians $J_F(\boldsymbol{u}_0^*)$ are symmetric. We provide more details on the relation between Eq. (6) and the usual gradient formula involving the energy function (Scellier & Bengio, 2017) in Appendix B. Intuitively, $\mathrm{d}_\beta \boldsymbol{u}^*$ can be seen as the replacement for $\boldsymbol{\delta}$ that the system can obtain using its own dynamics. When $J_F(\boldsymbol{u}_0^*) \neq J_F(\boldsymbol{u}_0^*)^{\top}$ (Fig. 1b), the asymmetric part of the Jacobian contributes an additional bias to the estimate of $\frac{\mathrm{d}\mathcal{L}}{\mathrm{d}\boldsymbol{\theta}}$. However, its contribution has never been studied independently because it is masked by the finite nudge $\beta$ in practice. It is worth noting that $\mathrm{d}_\beta \boldsymbol{u}^* = 0$ for feed forward networks due to the absence of feedback connections (Fig. 1c).

## 3 THEORETICAL RESULTS

### 3.1 GENERALIZED HOLOMORPHIC EP

To study individual bias contributions we build on Holomorphic EP (Laborieux & Zenke, 2022), which, assuming weight symmetry, allows computing unbiased gradients with finite nudges. As we will see, the property generalizes to asymmetric weights and $\mathrm{d}_\beta \boldsymbol{u}^*$ can be exactly computed with

finite teaching amplitude $|\beta| > 0$ for any complex differentiable (holomorphic) vector field $F$. We use the fact that for holomorphic functions, derivatives at a given point are equal to the Cauchy integral over a path around the point of interest $\beta = 0$ (Appel, 2007) (see Appendix A.1 for more details):

$$\left.\frac{\mathrm{d}\boldsymbol{u}^*}{\mathrm{d}\beta}\right|_{\beta=0} = \frac{1}{2\mathrm{i}\pi}\oint_\gamma \frac{\boldsymbol{u}_\beta^*}{\beta^2}\mathrm{d}\beta = \frac{1}{T|\beta|}\int_0^T \boldsymbol{u}_{\beta(t)}^* e^{-2\mathrm{i}\pi t/T}\mathrm{d}t. \tag{7}$$

Here we used the teaching signal $\beta(t) = |\beta|e^{2\mathrm{i}\pi t/T}, t \in [0, T]$ for the path $\gamma$, and performed the change of variable $\mathrm{d}\beta = \beta(t)\frac{2\mathrm{i}\pi}{T}\mathrm{d}t$, where $\mathrm{i} \in \mathbb{C}$ is the imaginary unit. Moreover, Eq. (7) assumes that the fixed point $\boldsymbol{u}^*$ is an implicit function of $\beta$ and well-defined on the entire path $\gamma$. In other words, we assume that the system is at equilibrium for all $t \in [0, T]$, which can be achieved when the timescale of the dynamical system is shorter than the one of the time-dependent nudge $\beta(t)$. In numerical simulations, we estimate the integral with a discrete number of $N$ points: $t/T = k/N, k \in [0, ..., N-1]$ (see Appendix E.1). Regardless of whether the dynamical system has a symmetric Jacobian at the free fixed point or not (Fig. 1), the exact value of $\mathrm{d}_\beta\boldsymbol{u}^*$ can be computed for a finite range of teaching amplitudes $|\beta|$ (Fig. 2a,b).

## 3.2 ESTIMATING GRADIENTS IN CONTINUOUS TIME

Although Eq. (7) implies that $\mathrm{d}_\beta\boldsymbol{u}^*$ is obtained at the end of one period of the teaching signal ($t = T$), it can also estimated continuously by letting the integral run over multiple oscillation cycles of the teaching signal $\beta(t)$ (Fig. 2 f):

$$\frac{1}{t|\beta|}\int_0^t \boldsymbol{u}_{\beta(\tau)}^* e^{-2\mathrm{i}\pi\tau/T}\mathrm{d}\tau \xrightarrow[t\to\infty]{} \left.\frac{\mathrm{d}\boldsymbol{u}^*}{\mathrm{d}\beta}\right|_{\beta=0}. \tag{8}$$

This relation is made obvious by splitting the integral over all the completed periods, such that the relative contribution of the remainder integral asymptotically approaches $0$ as $t \to \infty$ (see Appendix C.1). The advantage of this approach for physical systems is that $\mathrm{d}_\beta\boldsymbol{u}^*$ need not be accessed at the precise times $t = T$, thereby dispensing with the need for separate phases (Williams et al., 2023). Moreover, the free fixed point does not need to be evaluated to obtain $\mathrm{d}_\beta\boldsymbol{u}^*$. However, the pre-synaptic term $\frac{\partial F}{\partial \boldsymbol{\theta}}(\boldsymbol{u}_0^*)$ in Eq. (6) a priori still requires evaluating the free fixed point. We will see next that we can also estimate it continuously in time thanks to the Mean Value Theorem from complex analysis (Appel, 2007) (Fig. 2e), Appendix C.1:

$$\frac{1}{t}\int_0^t \frac{\partial F}{\partial \boldsymbol{\theta}}(\boldsymbol{u}_{\beta(\tau)}^*)\mathrm{d}\tau \xrightarrow[t\to\infty]{} \frac{\partial F}{\partial \boldsymbol{\theta}}(\boldsymbol{u}_0^*). \tag{9}$$

Overall, the gradient estimate $\tilde{\nabla}_{\boldsymbol{\theta}}$ in the time-continuous form corresponds to the average of pre-synaptic activation multiplied by the amplitude of the post-synaptic potential, consistent with models of STDP (Dan & Poo, 2004; Clopath & Gerstner, 2010). In the next section, we isolate the contribution to the bias brought by the asymmetry of the Jacobian $J_F(\boldsymbol{u}_0^*)$.

## 3.3 ISOLATING BIAS FROM JACOBIAN ASYMMETRY

We are now in the position to quantify the bias originating from asymmetric Jacobians in the computation of the neuronal error vector of hEP. By isolating $\frac{\partial F}{\partial \beta}$ and $\frac{\mathrm{d}\mathcal{L}}{\mathrm{d}\boldsymbol{u}_0^*}$ on both ends of Eq. (5), and because they are equal by Definiton 1, we see that both the neuronal error vectors of hEP and RBP are linked by the following relation:

$$\mathrm{d}_\beta\boldsymbol{u}^* = J_F(\boldsymbol{u}_0^*)^{-1}J_F(\boldsymbol{u}_0^*)^\top \boldsymbol{\delta}. \tag{10}$$

Thus, for asymmetric Jacobians, $\boldsymbol{\delta}$ and $\mathrm{d}_\beta\boldsymbol{u}^*$ are not aligned which introduces bias in the resulting gradient estimates. To further quantify this bias, we introduce $S$ and $A$ the symmetric and skew-symmetric parts of $J_F$, such that $J_F(\boldsymbol{u}_0^*) = S + A$ and $J_F(\boldsymbol{u}_0^*)^\top = S - A$. Then we can show that when the Frobenius norm $\left|\!\left|\!\left|S^{-1}A\right|\!\right|\!\right| \to 0$, $\mathrm{d}_\beta\boldsymbol{u}^*$ and $\boldsymbol{\delta}$ are linked by (see Appendix C.2 for a proof):

$$\mathrm{d}_\beta\boldsymbol{u}^* = \boldsymbol{\delta} - 2S^{-1}A\boldsymbol{\delta} + o(S^{-1}A\boldsymbol{\delta}). \tag{11}$$

This expression highlights the direct dependence of the bias on $A$ and further suggests a way to improve the Jacobian's symmetry, thus reducing the bias by directly decreasing the norm of $A$.

### 3.4 INCREASING FUNCTIONAL SYMMETRY THROUGH JACOBIAN HOMEOSTASIS.

To reduce the norm of $A$ we note that $\|A\|^2 = \text{Tr}(A^\top A) = \frac{1}{2}\text{Tr}(J_F^\top J_F) - \frac{1}{2}\text{Tr}(J_F^2)$, where the dependence of $J_F$ on $\boldsymbol{u}^*$ is omitted. Similar to work by Bai et al. (2021) on deep equilibrium models, we can use the classical Hutchinson trace estimator (Hutchinson, 1989) (see Appendix A.2) to obtain the following objective for minimizing Jacobian asymmetry and, therefore, improving "functional symmetry":

$$\mathcal{L}_{\text{homeo}} := \mathbb{E}_{\boldsymbol{\varepsilon} \sim \mathcal{N}(0,I)} \left[ \|J_F\boldsymbol{\varepsilon}\|^2 - \boldsymbol{\varepsilon}^\top J_F^2 \boldsymbol{\varepsilon} \right]. \tag{12}$$

If we break down the two terms present in this objective, we see that $\|J_F\boldsymbol{\varepsilon}\|^2$ should be minimized, which corresponds to making the network more robust to random perturbations by minimizing the Frobenius norm of the Jacobian. Interestingly, Stock et al. (2022) showed how a local heterosynaptic learning rule could optimize this objective locally through synaptic balancing (see Eqs. (12)–(14) in Stock et al. (2022)), whereby neurons try to balance their net inputs with their output activity. The second term in Eq. (12) comes with a minus sign and should therefore be maximized. It is, thus, akin to a reconstruction loss, similar to the training of feedback weights in target propagation (TP) (Meulemans et al., 2020; Ernoult et al., 2022; Lee et al., 2015), which can also be local (see e.g. Eq. (13) of Meulemans et al. (2021)), and reminiscent of predictive slow feature learning (Wiskott & Sejnowski, 2002; Lipshutz et al., 2020; Halvagal & Zenke, 2022). However, in our case *all* weights are trained to optimize $\mathcal{L}_{\text{homeo}}$, unlike TP approaches which only train feedback weights. Overall, $\mathcal{L}_{\text{homeo}}$ is a plausible objective for increasing and maintaining functional symmetry of the Jacobian in any dynamical system. Finally, it is important to note that while perfect weight symmetry implies functional symmetry, the converse is not true. For instance, Eq. (12) can be optimized in arbitrary systems without reciprocal connectivity. This distinction makes $\mathcal{L}_{\text{homeo}}$ more general than alternative strategies acting on weight symmetry directly (Kolen & Pollack, 1994; Akrout et al., 2019).

## 4 EXPERIMENTS

In the following experiments, we used the setting of convergent recurrent neural networks (Ernoult et al., 2019), as well as the linear readout for optimizing the cross-entropy loss (Laborieux et al., 2021) (see appendix E.2). Simulations were implemented in JAX (Bradbury et al., 2018) and Flax (Heek et al., 2020) and datasets obtained through the Tensorflow Datasets API (Abadi et al., 2015). Our code is available on GitHub[1] and hyperparameters can be found in Appendix E.3.

**Holomorphic EP matches automatic differentiation at computing $\mathrm{d}_\beta \boldsymbol{u}^*$.** We sought to understand the amount of bias due to the finite nudge $\beta$ when using different ways of estimating the neuronal error vector $\mathrm{d}_\beta \boldsymbol{u}^*$. To this end, we trained a two-hidden layer network with independent forward and backward connections (Fig. 1b) on the Fashion MNIST dataset (Xiao et al., 2017) using stochastic gradient descent for 50 epochs. To investigate the evolution of weight symmetry during training, the forward and backward connections were initialized symmetrically. We compared different ways of estimating the neuronal error vector $\mathrm{d}_\beta \boldsymbol{u}^*$. As the ideal "ceiling" case, we computed the ground truth using forward-mode automatic differentiation in JAX. Additionally, we computed the classic one-sided EP estimate (Scellier et al., 2018) denoted by "Classic" in Table 1, as well as the Cauchy integral estimate (Eq. 7) computed with various number $N$ of points (see Appendix E.1). In all cases we ran simulations with two teaching amplitudes $|\beta| = 0.05$ and $|\beta| = 0.5$. We report the final validation errors in Table 1. We observed that the bias of the nudge is already noticeable at low teaching amplitude when comparing to better estimates obtained using higher $N$ or the ground truth. The bias became more noticeable when the teaching amplitude was increased. The lowest average validation error was obtained with $N = 6$ points and $|\beta| = 0.5$, matching the ground truth within statistical uncertainty, and consistent with our theory. Finally, to test the potential of running hEP in continuous time, we implemented the continuous-time estimates of Eqs (8) and (9) where the average spanned five oscillation periods and $N = 4$ points were used. Crucially, there was no free phase in this setting. Despite longer numerical simulation times in this case (see Appendix E.4), we observed that the continuous-time estimate only suffered from moderate performance drop with respect to the ground truth $\mathrm{d}_\beta \boldsymbol{u}^*$ despite the complete absence of the free phase, thereby confirming our predictions.

---

[1]`https://github.com/Laborieux-Axel/generalized-holo-ep`

**Jacobian asymmetry leads to different learning dynamics.** To see whether and how Jacobian asymmetry influences training, we studied the dynamics of the neuronal error vectors $\mathrm{d}_\beta \boldsymbol{u}^*$, obtained through hEP, and $\boldsymbol{\delta}$ from RBP respectively. We used the same experimental setting as in the previous paragraph and trained neural networks with varying degrees of initial weight asymmetry. The degree of initial weight asymmetry was controlled by setting the backward weights $w_b$ as $w_b \leftarrow \sin(\alpha)w_b + \cos(\alpha)w_f^\top$, where $w_f$ are the forward weights. Throughout training, we recorded the evolution of the angle $\alpha$, and the cosine between both neuronal error vectors (Fig. 3).

We observed that networks trained with RBP all performed comparably in terms of loss (Fig. 3c) whereas their weight symmetry reduced quicker than in networks trained with hEP (Fig. 3a,b). Furthermore, RBP trained networks settled in a parameter regime in which the cosine between $\boldsymbol{\delta}$ and $\mathrm{d}_\beta \boldsymbol{u}^*$ was negative for all except the output layer (Fig. 3e,f). In contrast, the performance of networks trained with hEP strongly depended on the initial degree of symmetry (Fig. 3d), and although the cosine angle with RBP decreased during training, it remained positive throughout. These findings could suggest that learning with hEP alone reaches a parameter regime in which the cosine with $\boldsymbol{\delta}$ is too low to further decrease the loss while maintaining the current performance level. In summary, even networks that start with symmetric weights, lose this symmetry during training, which leads to increasing bias with respect to the ground-truth neuronal error vector and impaired task performance. In the next paragraph, we investigate whether and to what extent the homeostatic loss introduced above (cf Eq. (12)) can mitigate this problem.

Table 1: Validation error in % $\pm$ stddev on Fashion MNIST ($n = 5$) for different values of $|\beta|$.

| Teach. amp. | Classic | $N = 2$ | $N = 4$ | $N = 6$ | Continuous-time | True $\mathrm{d}_\beta \boldsymbol{u}^*$ |
|---|---|---|---|---|---|---|
| $|\beta| = 0.05$ | $15.8 \pm 0.6$ | $14.6 \pm 0.4$ | $14.7 \pm 0.7$ | $14.5 \pm 0.6$ | $15.8 \pm 0.8$ | $14.7 \pm 0.6$ |
| $|\beta| = 0.5$ | $38.4 \pm 6.2$ | $16.3 \pm 0.7$ | $14.8 \pm 0.8$ | $14.3 \pm 0.6$ | $17.3 \pm 1.0$ | |

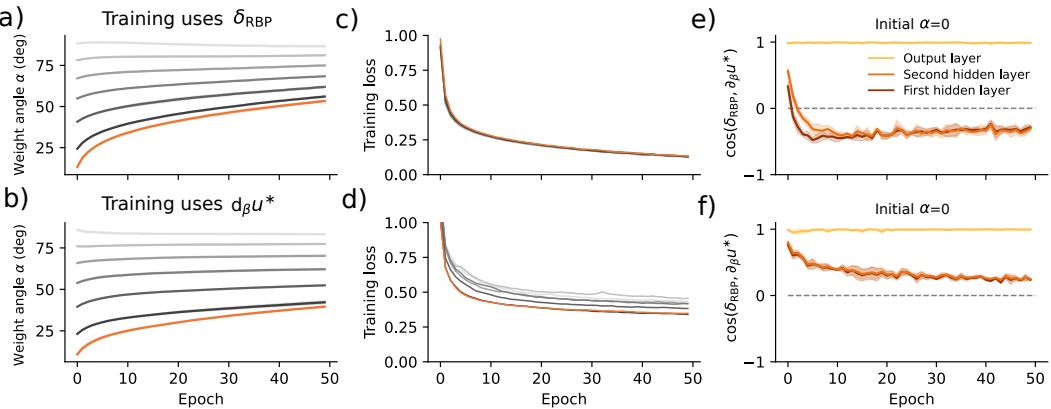

Figure 3: Comparison of training dynamics of a two-hidden-layer MLP on Fashion MNIST using RBP (top row) and generalized hEP (bottom row). **a,b)** Evolution of the angle between forward and backward connections during training on for varying initial angle. Training tends to reduce weight symmetry. **c,d)** Learning curves. The training loss is dependent on initial angle only for generalized hEP. **e,f)** Evolution of the cosine similarity between neuronal error vectors of RBP and hEP over training. Curves are averaged over three seeds and shaded areas denote $\pm 1$ stddev.

**Jacobian homeostasis improves functional symmetry and enables training on larger datasets.** To study the effect of the homeostatic loss (Eq. (12)), we added it scaled by the hyper parameter $\lambda_{\text{homeo}}$ to the cross entropy loss (see Appendix E.3). For all experiments we estimated the average in Eq. (12) using 5 stochastic realizations of a Gaussian noise $\varepsilon$ per sample in the mini-batch. With these settings, we trained the same network as in the previous paragraphs with the homeostatic loss

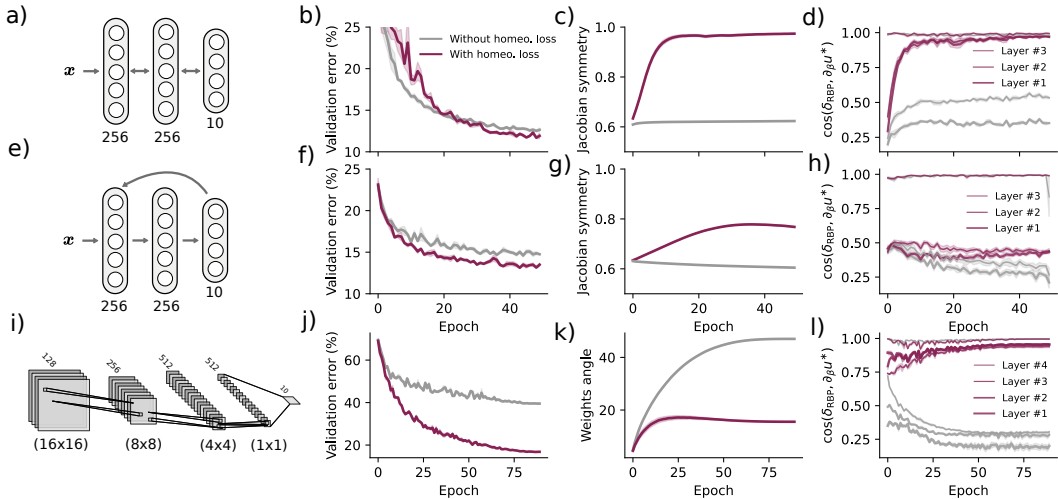

Figure 4: The homeostatic loss improves hEP training of arbitrary dynamical systems by acting directly on the Jacobian. Each row corresponds to a specific architecture-dataset pair. **a)** Multi-layer architecture where layers are reciprocally connected. Evolution of the validation error **b)**, Jacobian symmetry measure **c)**, and layer-wise cosine between neuronal error vectors of hEP and RBP **d)** during training of Fashion MNIST. **e-h)** Same plots as **a-d** for an architecture where the output layer directly feeds back to the first layer. **i)** Recurrent convolutional architecture used on CIFAR-10. Evolution of validation error **j)**, angle between weights **k)** and layer-wise cosine between the neuronal error vectors of hEP and RBP **l)**. Curves are averaged over five random seeds for Fashion MNIST and three for CIFAR-10 and shaded areas denote $\pm$ 1 standard error.

on Fashion MNIST (Fig. 4a) and found a small reduction in validation error. We further observed that the added homeostatic loss increased the symmetry of the Jacobian, as defined by the symmetry measure $\|S\|/(\|S\| + \|A\|)$ (Section 3), over the course of training. In line with this observation, the alignment between $\boldsymbol{\delta}$ and $\mathrm{d}_\beta \boldsymbol{u}^*$ (Fig. 4c-d) also increased for each layer.

Because the homeostatic loss is defined on the Jacobian instead of the weights, we hypothesized that its effect may be more general than merely increasing weight symmetry. To test this idea, we introduced a network architecture without feedback connections between adjacent layers, thereby precluding the possibility for symmetric feedback. Specifically, we trained a multi-layer architecture in which the output units fed back to the first hidden layer directly (Fig. 4e) (Kohan et al., 2018; 2023; Dellaferrera & Kreiman, 2022). After training this antisymmetric architecture, we observed the same key effects on the validation error, Jacobian symmetry, and error vector alignment (Fig. 4g,h) as in the reciprocally connected network, thereby confirming our hypothesis that the homeostatic loss is more general than weight alignment (Kolen & Pollack, 1994). In addition, we show in Appendix D that the homeostatic loss generalize to predictive coding networks (Whittington & Bogacz, 2019).

Since, the improvement in validation error was measurable, but small in the network architectures we studied (Fig. 4b,f), we wondered whether the positive effect of the homeostatic loss would be more pronounced on more challenging tasks. To this end, we extended a recurrent convolutional architecture (Fig. 4i; Laborieux & Zenke, 2022) to asymmetric feedback weights (see Appendix E.2 for further details). We trained this architecture using generalized hEP on CIFAR-10, CIFAR-100 (Krizhevsky, 2009) vision benchmarks both with and without homeostatic loss.

Without the homeostatic loss, the network trained on CIFAR-10 reached 60.4% validation accuracy (Table 2). With the homeostatic loss, the performance increased to 84.3 (Table 2), with only approximate weight symmetry (Fig. 4k). Strikingly, this is only a reduction by 4.3% in accuracy in comparison to the symmetric architecture (Laborieux & Zenke, 2022). When using $N = 2$ instead of the ground truth for estimating $\mathrm{d}_\beta \boldsymbol{u}^*$, we observed an additional drop by 2.9% points due to the finite nudge bias. We also noticed that the homeostatic loss has no measurable effect on performance when training the asymmetric network with RBP (Table 2), suggesting that the homeostatic loss is only beneficial while not restricting model performance. Finally, we reproduced similar findings on

Table 2: Validation accuracy of asymmetric networks trained with generalized hEP (ours) and our implementation of RBP. All values are averages ($n = 3$) $\pm$ stddev.

| | CIFAR-10 | CIFAR-100 | | ImageNet $32 \times 32$ | |
|---|---|---|---|---|---|
| | Top-1 (%) | Top-1 (%) | Top-5 (%) | Top-1 (%) | Top-5 (%) |
| hEP w/o $\mathcal{L}_{\text{homeo}}$ | $60.4 \pm 0.4$ | $35.2 \pm 0.3$ | $64.4 \pm 0.5$ | – | – |
| hEP$_{N=2,|\beta|=1}$ | $81.4 \pm 0.1$ | $51.1 \pm 0.8$ | $79.2 \pm 0.4$ | – | – |
| hEP (True $\mathrm{d}_\beta \boldsymbol{u}^*$) | $84.3 \pm 0.1$ | $53.8 \pm 0.8$ | $81.0 \pm 0.3$ | $31.4 \pm 0.1$ | $55.2 \pm 0.1$ |
| RBP | $87.8 \pm 0.3$ | $60.8 \pm 0.2$ | $84.4 \pm 0.1$ | – | – |
| RBP w/o $\mathcal{L}_{\text{homeo}}$ | $87.7 \pm 0.2$ | $61.0 \pm 0.2$ | $85.2 \pm 0.3$ | – | – |
| Sym. hEP$_{N=2,|\beta|=1}$ | $88.6 \pm 0.2$ | $61.6 \pm 0.1$ | $86.0 \pm 0.1$ | $36.5 \pm 0.3$ | $60.8 \pm 0.4$ |

CIFAR-100 and ImageNet $32 \times 32$, with a remaining 5% gap on Top-5 validation accuracy from the perfectly symmetric architecture. Together these findings suggest that the addition of homeostatic objectives is increasingly important for training on larger datasets and further highlights the need additional homeostatic processes in the brain (Zenke et al., 2017; Stock et al., 2022).

## 5 Discussion

We have studied a generalized form of EP in the absence of perfect weight symmetry for finite nudge amplitudes. We found that relaxing the strong algorithmic assumptions underlying EP rapidly degraded its ability to estimate gradients in larger networks and on real-world tasks. We identified two main sources of bias in generalized EP (Scellier et al., 2018) with adverse effect on performance: The finite nudge amplitude and the Jacobian asymmetry. Further, we illustrate how both issues can be overcome through oscillations by combining holomorphic EP (Laborieux & Zenke, 2022) with a new form of Jacobian homeostasis that encourages functional symmetry, but without imposing strict weight symmetry. Finally, we show that our strategy allows training deep dynamical networks without perfect weight symmetry on ImageNet $32 \times 32$ with only a small gap in performance to ideal symmetric networks.

The role of weight asymmetry for computing gradients has received a lot of attention (Kolen & Pollack, 1994; Lillicrap et al., 2016; Launay et al., 2020; Nøkland, 2016; Payeur et al., 2021; Greedy et al., 2022). Our work corroborates previous findings showing that learning with fully asymmetric weights results in poor alignment with back propagation which limits it to small datasets. Reminiscent of work on Feedback Alignment (Lillicrap et al., 2016), training becomes more challenging in deeper networks and on larger datasets (Xiao et al., 2018; Liao et al., 2016; Bartunov et al., 2018). Although our findings do not allow completely dispensing with the weight symmetry requirement, they are more general since Jacobian symmetry and weight symmetry are not the same, a realization that may prove useful for future algorithmic developments.

Nevertheless, several questions pertaining to plausibility remain open. For instance, removing the bias with hEP requires the neurons to oscillate in the complex plane, which restricts its use to oscillation-based learning and makes its biological interpretation challenging. It could be implemented through phase coding (Frady & Sommer, 2019; Bybee et al., 2022), which would require an additional fast carrier frequency, suggesting potential links to the beta and gamma rhythm in neurobiology (Engel et al., 2001). Another limitation of EP is the convergence to an equilibrium. This requirement makes EP costly on digital computers, which simulate the substrate physics, whereas analog substrates could achieve this relaxation following the laws of physics (Yi et al., 2023; Kendall et al., 2020).

In summary, our work further bridges the gap between EP's assumptions and constraints found in physical neural networks, and opens new avenues for understanding and designing oscillation-based learning algorithms for power efficient learning systems.

## ACKNOWLEDGMENTS

We thank all members of the Zenke Lab for comments and discussions. We thank Maxence Ernoult, Nicolas Zucchet, Jack Kendall and Benjamin Scellier for helpful feedback. This project was supported by the Swiss National Science Foundation [grant numbers PCEFP3_202981 and TMPFP3_210282], EU's Horizon Europe Research and Innovation Programme (CONVOLVE, grant agreement number 101070374) funded through SERI (ref 1131-52302), and the Novartis Research Foundation. The authors declare no competing interests.

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

# A MATHEMATICAL BACKGROUND

## A.1 COMPLEX ANALYSIS

Here we provide intuitions and the minimal theoretical background on complex analysis to allow appreciating the results presented in Sections 3.1 and 3.2. We refer the interested reader to (Appel, 2007) for a more complete introduction.

The notion of complex differentiability in a point $z_0 \in \mathbb{C}$ is defined for a function $f : z \in \mathbb{C} \mapsto f(z) \in \mathbb{C}$ by the existence of the limit:

$$\lim_{z \to z_0} \frac{f(z) - f(z_0)}{z - z_0}. \tag{13}$$

The limit is written $f'(z_0)$, and the function $f$ is said to be "holomorphic" in $z_0$. Although the definition is similar to $\mathbb{R}$-differentiability, the fact that the limit is taken with a complex number that can go to $z_0$ from all directions in the complex plane makes the variation of $f$ more "rigid". However, most usual functions on real numbers can be extended to complex inputs and outputs and are holomorphic, such as the exponential, the sigmoid, all polynomials, trigonometric functions, softmax, and logarithm. Non-holomorphic functions include functions that, for instance, use the absolute value or min/max operators.

In exchange for the rigidness of holomorphic functions, one obtains the possibility to express their derivatives at any order through integrals as described by the Cauchy formulas.

$$f^{(n)}(z_0) = \frac{n!}{2\mathrm{i}\pi} \oint_\gamma \frac{f(z)}{(z - z_0)^{n+1}} \mathrm{d}z, \tag{14}$$

where $n!$ is the factorial product of integers up to $n$, i is the imaginary unit, and the integral is taken over a path $\gamma \subset \mathbb{C}$ going around $z_0$ once and counterclockwise, and provided that $f$ is holomorphic on a set that includes the path. For instance, for $n = 0$ and the path $\gamma : \theta \in [-\pi, \pi] \mapsto z_0 + re^{\mathrm{i}\theta}$, doing the change of variable $\mathrm{d}z = r\mathrm{i}e^{\mathrm{i}\theta}\mathrm{d}\theta$ yields the mean value property linking the value of $f$ in $z_0$ to its variation on a circle of radius $r$ going around $z_0$:

$$f(z_0) = \frac{1}{2\mathrm{i}\pi} \int_{-\pi}^{\pi} \frac{f(z_0 + re^{\mathrm{i}\theta})}{z_0 + re^{\mathrm{i}\theta} - z_0} r\mathrm{i}e^{\mathrm{i}\theta}\mathrm{d}\theta,$$

$$f(z_0) = \frac{1}{2\pi} \int_{-\pi}^{\pi} f(z_0 + re^{\mathrm{i}\theta})\mathrm{d}\theta. \tag{15}$$

And $f'(z_0)$ can be computed by plugging $n = 1$ in Eq. (14), and the same change of variable:

$$f'(z_0) = \frac{1}{2\mathrm{i}\pi} \oint_\gamma \frac{f(z)}{(z - z_0)^2} \mathrm{d}z,$$

$$f'(z_0) = \frac{1}{2\mathrm{i}\pi} \int_{-\pi}^{\pi} \frac{f(z_0 + re^{\mathrm{i}\theta})}{(z_0 + re^{\mathrm{i}\theta} - z_0)^2} r\mathrm{i}e^{\mathrm{i}\theta}\mathrm{d}\theta,$$

$$f'(z_0) = \frac{1}{2\pi r} \int_{-\pi}^{\pi} f(z_0 + re^{\mathrm{i}\theta})e^{-\mathrm{i}\theta}\mathrm{d}\theta. \tag{16}$$

Importantly, these formulas are integrals over a non-vanishing radius $r > 0$ where the integrands only involve the function $f$, which is a promising conceptual step toward biological plausiblity and hardware design since integrals are easier to implement in noisy substrates than finite differences.

## A.2 HUTCHINSON TRACE ESTIMATOR

The Hutchinson trace estimator (Hutchinson, 1989) refers to the following "trick" to estimate the trace of a square matrix $M = [m_{ij}]$ : suppose we have a random vector $\varepsilon \in \mathbb{R}^n$ such that $\mathbb{E}[\varepsilon\varepsilon^\top] = I$,

where $I$ is the identity matrix. Then we have:

$$
\begin{aligned}
\mathbb{E}[\boldsymbol{\varepsilon}^\top M \boldsymbol{\varepsilon}] &= \mathbb{E}\left[\sum_{ij} m_{ij}\varepsilon_i\varepsilon_j\right] \\
&= \sum_{ij} m_{ij}\mathbb{E}\left[\varepsilon_i\varepsilon_j\right] \\
&= \sum_{ij} m_{ij}I_{ij} \\
&= \sum_i m_{ii} = \mathrm{Tr}(M).
\end{aligned} \tag{17}
$$

Since the Frobenius norm can be expressed with the trace operator, it can be estimated in the same way. For instance, the Frobenius norm of the asymmetric part $A$ of the Jacobian $J_F$, $A = \frac{1}{2}(J_F - J_F^\top)$ is given by:

$$
\begin{aligned}
\||A\||^2 &= \mathrm{Tr}(A^\top A) \\
&= \frac{1}{4}\mathrm{Tr}\left((J_F^\top - J_F)(J_F - J_F^\top)\right) \\
&= \frac{1}{4}\mathrm{Tr}\left(J_F^\top J_F - J_F^\top J_F^\top - J_F J_F + J_F J_F^\top\right) \\
&= \frac{1}{2}\mathrm{Tr}(J_F^\top J_F) - \frac{1}{2}\mathrm{Tr}(J_F^2) \\
&= \frac{1}{2}\mathbb{E}[\boldsymbol{\varepsilon}^\top J_F^\top J_F \boldsymbol{\varepsilon}] - \frac{1}{2}\mathbb{E}[\boldsymbol{\varepsilon}^\top J_F^2 \boldsymbol{\varepsilon}] \quad \text{using Eq. (17)} \\
&= \frac{1}{2}\mathbb{E}\left[\|J_F\boldsymbol{\varepsilon}\|^2 - \boldsymbol{\varepsilon}^\top J_F^2 \boldsymbol{\varepsilon}\right].
\end{aligned} \tag{18}
$$

To obtain the fourth line, we used the linear property of the trace operator, the fact that the trace is unchanged by the matrix transposition, and fact that $\mathrm{Tr}(M_1 M_2) = \mathrm{Tr}(M_2 M_1)$ for any two matrices $M_1$ and $M_2$, which is a consequence of the symmetry of the Frobenius inner product. The final quantity in Eq. (18) can be estimated efficiently with vector-Jacobian product routines available in automatic differentiation frameworks such as JAX or PyTorch, without instantiating the full Jacobian matrix.

## B  RELATION BETWEEN THE ENERGY AND VECTOR FIELD GRADIENT FORMULAS

Here we briefly provide an intuition for how the EP gradient formula obtained with the energy function formalism (Scellier & Bengio, 2017) relates to the EP gradient formula obtained directly with the vector field (Eq. (6), Scellier et al. (2018)). Although the formulas seem different, they are in fact equivalent. The main takeaway is that the connections $w_{ij}$ and $w_{ji}$ are fused into a single parameter in the energy function formalism, whereas the vector field formalism distinguishes the two. For a continuous Hopfield model with two-body interaction terms of the form $-\sigma(u_i)w_{ij}\sigma(u_j)$, the energy-based (EB) gradient of the loss with respect to $w_{ij}$ is (Scellier & Bengio, 2017):

$$
\left.\frac{\mathrm{d}\mathcal{L}}{\mathrm{d}w_{ij}}\right|_{\mathrm{EB}} = \left.\frac{\mathrm{d}\sigma(u_i^*)\sigma(u_j^*)}{\mathrm{d}\beta}\right|_{\beta=0}.
$$

If we use the product rule of derivatives, we obtain:

$$
\left.\frac{\mathrm{d}\mathcal{L}}{\mathrm{d}w_{ij}}\right|_{\mathrm{EB}} = \sigma(u_i^*)\sigma'(u_j^*) \left.\frac{\mathrm{d}u_j^*}{\mathrm{d}\beta}\right|_{\beta=0} + \sigma(u_j^*)\sigma'(u_i^*) \left.\frac{\mathrm{d}u_i^*}{\mathrm{d}\beta}\right|_{\beta=0}.
$$

Both terms in the sum corresponds respectively to the vector field (VF) gradient formula for $w_{ij}$ and $w_{ji}$. Here $\frac{\partial F}{\partial w_{ij}} = \sigma(u_i^*)\sigma'(u_j^*)$. Therefore, we have:

$$
\left.\frac{\mathrm{d}\mathcal{L}}{\mathrm{d}w_{ij}}\right|_{\mathrm{EB}} = \left.\frac{\mathrm{d}\mathcal{L}}{\mathrm{d}w_{ij}}\right|_{\mathrm{VF}} + \left.\frac{\mathrm{d}\mathcal{L}}{\mathrm{d}w_{ji}}\right|_{\mathrm{VF}}.
$$

## C  THEORETICAL PROOFS

### C.1  CONVERGENCE OF THE CONTINUOUS-TIME ESTIMATE

We recall the limit:

$$\frac{1}{t|\beta|}\int_0^t \boldsymbol{u}^*_{\beta(\tau)}e^{-2\mathrm{i}\pi\tau/T}\mathrm{d}\tau \underset{t\to\infty}{\longrightarrow} \frac{\mathrm{d}\boldsymbol{u}^*}{\mathrm{d}\beta}\bigg|_{\beta=0}.$$

We split the running time $t$ into an integer $k$ amount of periods $t = kT + t'$. Then by splitting the integral we have:

$$
\begin{aligned}
\frac{1}{t|\beta|}\int_0^t \boldsymbol{u}^*_{\beta(\tau)}e^{-2\mathrm{i}\pi\tau/T}\mathrm{d}\tau &= \frac{1}{t|\beta|}\int_0^{kT+t'} \boldsymbol{u}^*_{\beta(\tau)}e^{-2\mathrm{i}\pi\tau/T}\mathrm{d}\tau \\
&= \frac{1}{t|\beta|}\left(k\int_0^T \boldsymbol{u}^*_{\beta(\tau)}e^{-2\mathrm{i}\pi\tau/T}\mathrm{d}\tau + \int_0^{t'} \boldsymbol{u}^*_{\beta(\tau)}e^{-2\mathrm{i}\pi\tau/T}\mathrm{d}\tau\right) \\
&= \frac{1}{t|\beta|}\left(kT|\beta|\frac{\mathrm{d}\boldsymbol{u}^*}{\mathrm{d}\beta}\bigg|_{\beta=0} + \int_0^{t'} \boldsymbol{u}^*_{\beta(\tau)}e^{-2\mathrm{i}\pi\tau/T}\mathrm{d}\tau\right) \\
&= \frac{t-t'}{t}\frac{\mathrm{d}\boldsymbol{u}^*}{\mathrm{d}\beta}\bigg|_{\beta=0} + \frac{1}{t|\beta|}\int_0^{t'} \boldsymbol{u}^*_{\beta(\tau)}e^{-2\mathrm{i}\pi\tau/T}\mathrm{d}\tau.
\end{aligned}
\tag{19}
$$

Since $0 < t' < T$, the first term converges toward $\mathrm{d}_\beta\boldsymbol{u}^*$, and the second term is a bounded integral divided by $t$, and therefore goes to 0 as $t \to \infty$.

For the pre-synaptic term we recall the statement:

$$\frac{1}{t}\int_0^t \frac{\partial F}{\partial\boldsymbol{\theta}}(\boldsymbol{u}^*_{\beta(\tau)})\mathrm{d}\tau \underset{t\to\infty}{\longrightarrow} \frac{\partial F}{\partial\boldsymbol{\theta}}(\boldsymbol{u}^*_0).$$

The proof is the same as above, once we have shown that:

$$\frac{1}{T}\int_0^T \frac{\partial F}{\partial\boldsymbol{\theta}}(\boldsymbol{u}^*_{\beta(\tau)})\mathrm{d}\tau = \frac{\partial F}{\partial\boldsymbol{\theta}}(\boldsymbol{u}^*_0).\tag{20}$$

By applying the Mean Value Theorem (Eq. (15)) (Appel, 2007) to the function $\varphi : \beta \in \mathcal{D}_{\boldsymbol{u}^*} \mapsto \frac{\partial F}{\partial\boldsymbol{\theta}}(\boldsymbol{u}^*_\beta)$ in $\beta = 0$, we have that:

$$\frac{\partial F}{\partial\boldsymbol{\theta}}(\boldsymbol{u}^*_0) = \frac{1}{2\pi}\int_0^{2\pi} \frac{\partial F}{\partial\boldsymbol{\theta}}(\boldsymbol{u}^*_{re^{\mathrm{i}\alpha}})\,\mathrm{d}\alpha,$$

where $r$ is a radius such that the circle is contained in the domain of definition $\mathcal{D}_{\boldsymbol{u}^*}$ of $\varphi$. Then we apply the equation in $r = |\beta|$ and do the change of variable $\alpha = 2\pi\tau/T$, which gives Eq. (20).

### C.2  BIAS TERM FROM JACOBIAN ASYMMETRY

We recall that $\mathrm{d}_\beta\boldsymbol{u}^*$ and $\boldsymbol{\delta}$ are linked by:

$$\mathrm{d}_\beta\boldsymbol{u}^* = J_F(\boldsymbol{u}^*_0)^{-1}J_F(\boldsymbol{u}^*_0)^\top\boldsymbol{\delta}.\tag{21}$$

Let $S$ and $A$ be the symmetric and skew-symmetric parts of $J_F$, such that $S = (J_F(\boldsymbol{u}^*_0) + J_F(\boldsymbol{u}^*_0)^\top)/2$ and $A = (J_F(\boldsymbol{u}^*_0) - J_F(\boldsymbol{u}^*_0)^\top)/2$. Since $J_F(\boldsymbol{u}^*_0)$ is assumed invertible and $|\!|\!|A|\!|\!|$

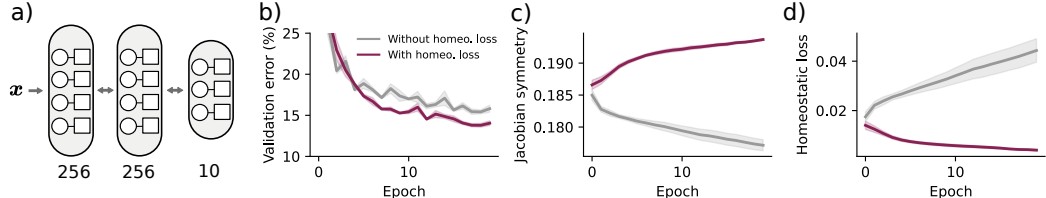

Figure 5: The same experiment as Fig. 4, but run on a Predictive Coding Network (Whittington & Bogacz, 2019). **a)** Architecture where each layer consists of value neurons (circles) and error neurons (squares). Evolution during training of the validation error **b)**, Jacobian symmetry measure **c)**, and Homeostatic loss **d)**.

is assumed small, we have that $S$ is invertible and:

$$
\begin{aligned}
J_F(\boldsymbol{u}_0^*)^{-1} J_F(\boldsymbol{u}_0^*)^\top &= (S+A)^{-1}(S-A) \\
&= (I+S^{-1}A)^{-1}S^{-1}(S-A) \\
&= (I+S^{-1}A)^{-1}(I-S^{-1}A) \\
&= \left(\sum_{k=0}^{\infty}(-1)^k(S^{-1}A)^k\right)(I-S^{-1}A) \\
&= \sum_{k=0}^{\infty}(-1)^k(S^{-1}A)^k - \sum_{k=0}^{\infty}(-1)^k(S^{-1}A)^{k+1} \\
&= \sum_{k=0}^{\infty}(-1)^k(S^{-1}A)^k + \sum_{k=1}^{\infty}(-1)^k(S^{-1}A)^k \\
&= I - 2S^{-1}A + 2\sum_{k=2}^{\infty}(-1)^k(S^{-1}A)^k.
\end{aligned}
$$

We used the Neumann Series to compute the inverse $(I+S^{-1}A)^{-1}$, which converges given the assumption $\left\|\left|S^{-1}A\right|\right\| \to 0$. From Eq (21) we can write:

$$
\mathrm{d}_\beta \boldsymbol{u}^* = \boldsymbol{\delta} - 2S^{-1}A\boldsymbol{\delta} + o(S^{-1}A\boldsymbol{\delta})
$$

## D   APPLICATION TO PREDICTIVE CODING NETWORKS

To demonstrate that our approach of regularizing the Jacobian to improve its symmetry is not restricted to the networks trained in the main manuscript, and generalize to other dynamical systems, we run an additional training experiment on a different model architecture, namely predictive coding networks (PCNs) (Whittington & Bogacz, 2019).

The key feature of the predictive coding network architecture is the presence in each layer $l$ of explicit error neurons $\boldsymbol{\epsilon}_l$ in addition to value neurons $\boldsymbol{u}_l$ (Fig. 5). The evolution of the error and value neurons are given by:

$$
\boldsymbol{\epsilon}_l = \boldsymbol{u}_l - \mathbf{w}_l^f \sigma(\boldsymbol{u}_{l-1}),
$$

$$
\frac{\mathrm{d}\boldsymbol{u}_l}{\mathrm{d}t} = -\boldsymbol{\epsilon}_l + \sigma'(\boldsymbol{u}_l) \odot \mathbf{w}_{l+1}^b \boldsymbol{\epsilon}_{l+1},
$$

where $\odot$ denotes the element-wise product, and the superscripts $f$ and $b$ denote forward and backward weights respectively. We train such a network with generalized hEP on Fashion MNIST with and without the homeostatic loss, and observe the same behavior as in Fig. 4 that optimizing the homeostatic loss for the PCN improves the symmetry of the Jacobian of the network, which translates into better validation error (Fig. 5). The hyperparameters used in this experiment are reported in Table 3.

# E    SIMULATION DETAILS

## E.1    NUMERICAL ESTIMATE OF THE CAUCHY INTEGRAL

For evaluating the integral in Eq. (7), we set an integer number of $N > 1$ values of $\beta$:

$$\beta_k = |\beta|e^{2i\pi k/N}, \quad k \in [0, 1, ..., N-1].$$

The integral is then approximated by:

$$\frac{1}{T|\beta|}\int_0^T \boldsymbol{u}^*_{\beta(\tau)}e^{-2i\pi\tau/T}\mathrm{d}\tau \approx \frac{1}{N|\beta|}\sum_{k=0}^N \boldsymbol{u}^*_{\beta_k}e^{-2i\pi k/N}. \tag{22}$$

The leading term of the bias when using $N$ points is $|\beta|^N/(N+1)!$, where ! is the factorial product. However, unless specified we used forward mode automatic differentiation to compute the ground truth $\mathrm{d}_\beta\boldsymbol{u}^*$ without bias.

## E.2    ARCHITECTURE AND DYNAMICS

**Multi-layer networks.**    The disrete dynamics of the multi-layer networks of experiments in Figs. 2, 3, 4 on Fashion MNIST read:

$$\begin{cases} \boldsymbol{u}_1 & \leftarrow & \mathbf{w}_{in}\boldsymbol{x} + \sum_{l' \text{ to } 1}\mathbf{w}_{1,l'}\sigma(\boldsymbol{u}_{l'}) + \mathbf{b}_1, \\ \boldsymbol{u}_l & \leftarrow & \sum_{l' \text{ to } l}\mathbf{w}_{l,l'}\sigma(\boldsymbol{u}_{l'}) + \mathbf{b}_l, \\ \boldsymbol{u}_L & \leftarrow & \mathbf{w}^f_{L,L-1}\sigma(\boldsymbol{u}_{L-1}) + \mathbf{b}_L + \beta\mathbf{w}^\top_{ro}\left(\mathbf{y} - \mathrm{softmax}\left(\mathbf{w}_{ro}\sigma(\boldsymbol{u}_L) + \mathbf{b}_{ro}\right)\right), \end{cases}$$

where $\mathbf{w}_{l,l'}$ is the weight matrix going from layer $l'$ to $l$, and $\mathbf{b}_l$ the bias of layer $l$. $(\mathbf{w}_{ro}, \mathbf{b}_{ro})$ is a $10 \times 10$ readout layer of the parameterized cross-entropy loss of Laborieux et al. (2021). In particular, it does not belong to the network connections since it does not influence the free dynamics (when $\beta = 0$). The activation $\sigma$ is the shifted sigmoid $x \mapsto 1/(1 + e^{-4x+2})$.

**Convolutional networks.**    For the convolutional neural network experiments, the discrete dynamics read:

$$\begin{cases} \boldsymbol{u}_1 & \leftarrow & \mathcal{P}(\mathbf{w}^f_1 * \boldsymbol{x}) + \tilde{\mathcal{P}}(\boldsymbol{u}_1, \mathbf{w}^b_2, \boldsymbol{u}_2) + \mathbf{b}_1, \\ \boldsymbol{u}_2 & \leftarrow & \mathcal{P}(\mathbf{w}^f_2 * \sigma(\boldsymbol{u}_1)) + \tilde{\mathcal{P}}(\boldsymbol{u}_2, \mathbf{w}^b_3, \boldsymbol{u}_3) + \mathbf{b}_2, \\ \boldsymbol{u}_3 & \leftarrow & \mathcal{P}(\mathbf{w}^f_3 * \sigma(\boldsymbol{u}_2)) + \tilde{\mathcal{P}}(\boldsymbol{u}_3, \mathbf{w}^b_4, \boldsymbol{u}_4) + \mathbf{b}_3, \\ \boldsymbol{u}_4 & \leftarrow & \mathcal{P}(\mathbf{w}^f_4 * \sigma(\boldsymbol{u}_3)) + \mathbf{b}_4 + \beta\mathbf{w}^\top_{ro}\left(\mathbf{y} - \mathrm{softmax}\left(\mathbf{w}_{ro}\sigma(\boldsymbol{u}_L) + \mathbf{b}_{ro}\right)\right). \end{cases}$$

Where the superscripts $f, b$ respectively mean forward and backward weights. $\mathcal{P}$ denotes Softmax pooling (Stergiou et al., 2021). The backward convolutional module going from $\boldsymbol{u}_{l+1}$ to $\boldsymbol{u}_l$ is defined as:

$$\tilde{\mathcal{P}}(\boldsymbol{u}_l, \mathbf{w}^b, \boldsymbol{u}_{l+1}) := \frac{\partial}{\partial\sigma(\boldsymbol{u}_l)}\left[\boldsymbol{u}_{l+1} \cdot \mathcal{P}(\mathbf{w}^b * \sigma(\boldsymbol{u}_l))\right].$$

The activation $\sigma$ is a sigmoid-weighted linear unit (Elfwing et al., 2018) defined by:

$$\sigma(x) := \left(\frac{x}{2}\right)\frac{1}{1 + e^{-x}} + \left(1 - \frac{x}{2}\right)\frac{1}{1 + e^{-x+2}}.$$

The shapes of $\boldsymbol{u}_1, \boldsymbol{u}_2, \boldsymbol{u}_3, \boldsymbol{u}_4$ are respectively $(16, 16, 128), (8, 8, 256), (4, 4, 512), (1, 1, 512)$. The weights are all $3 \times 3$ kernel with no strides. The paddings are all 'same' except for the last layer. The Softmax pooling has stride 2 and window-size $2 \times 2$. It is defined by (Stergiou et al., 2021):

$$y = \sum_{i\in\mathbf{R}}\left(\frac{e^{x_i}}{\sum_{j\in\mathbf{R}}e^{x_j}}\right)x_i,$$

where $\mathbf{R}$ is the $2 \times 2$ window.

### E.3 HYPERPARAMETERS

**Choice of hyperparameters.** The learning rate was the main hyperparameter searched for by coarse grid search. The number of time steps was tuned such that the networks have enough time to equilibrate by measuring the norm of the vector field. The coefficient for the homeostatic loss was searched by grid search over the range of 0.1, 1.0, and 10.0. Hyperparameters for the CIFAR-10/100 and ImageNet $32 \times 32$ were chosen based on those reported in the literature for similar experiments (Laborieux et al., 2021; Laborieux & Zenke, 2022).

Table 3: Hyperparameters used for the Fashion MNIST training experiments.

| Hyperparameter | Fig. 3, Table 1, Fig. 4e-h) | Fig. 4a-d) | Fig. 5 |
|---|---|---|---|
| Batch size | 50 | 50 | 50 |
| Optimizer | SGD | Adam | SGD |
| Loss | X-ent. | X-ent. | Squared Error |
| Learning rate | 1e-2 | 1e-4 | 1e-2 |
| Momentum | 0.9 | – | 0.9 |
| Epochs | 50 | 50 | 20 |
| $\lambda_{\text{homeo}}$ | 0.1 (Fig. 4e-h) | 1.0 | 1.0 |
| $T_{\text{free}}$ | 150 | 150 | 50 |
| $T_{\text{nudge}}$ | 20 | 20 | 20 |

In the case of continuous-time estimates (Table 1, Eqs. (8), (9)), we used 60 time steps for each of the $N = 4$ $\beta$ values, and five periods, for a total to 1200 time steps per batch.

Table 4: Hyperparameters used for the CIFAR-10, CIFAR-100 and ImageNet $32 \times 32$ training experiments.

| Hyperparameter | Fig. 4i-l) | Table 2 CIFAR-(10/100) | Table 2 ImageNet $32 \times 32$ |
|---|---|---|---|
| Batch size | 256 | 256 | 256 |
| Optimizer | SGD | SGD | SGD |
| Learning rate | 5e-3 | 7e-3 | 5e-3 |
| Weight decay | 1e-4 | 1e-4 | 1e-4 |
| Momentum | 0.9 | 0.9 | 0.9 |
| Epochs | 90 | 150 | 90 |
| $\lambda_{\text{homeo}}$ | 10 or 0 | 15 | 10 |
| $T_{\text{free}}$ | 250 | 250 | 250 |
| $T_{\text{nudge}}$ | 40 | 40 | 40 |

\* Learning rates were scaled layer-wise by [25, 15, 10, 8, 5] and decayed with cosine schedule (Loshchilov & Hutter, 2017).

### E.4 HARDWARE AND SIMULATION TIMES

Simulations on Fashion MNIST were run on single RTX 5000 GPU, for 10mins per run and 1 hour for the continuous-time estimate. The convolutional network simulations were run on an in-house cluster consisting of 5 nodes with 4 v100 NVIDIA GPUs each, one node with 4 A100 NVIDIA GPUs, and one node with 8 A40 NVIDIA GPUs. Runs on CIFAR-10/100 took about 8 hours each, and the Imagenet $32 \times 32$ runs took 72 hours each. Each run was parallelized over 4 or 8 GPUs by splitting the batch. The free phase with the convolutional architecture was done in half precision, which provided a $2\times$ speed up compared to using full precision.

