# OpenReview forum: "Improving equilibrium propagation without weight symmetry through Jacobian homeostasis"
_ICLR.cc/2024/Conference — ICLR 2024 poster_

### Official Review · Reviewer_2L2m · 2023-10-28

**Soundness:** 3 good
**Presentation:** 3 good
**Contribution:** 2 fair
**Rating:** 5
**Confidence:** 3

**Summary:**

The authors suggest a modification to the holomorphic equilibrium propagation algorithm, that helps it deal with cases where the Jacobian is not symmetric, by adding a term which penalizes asymmetry of the Jacobian. They show this approach outperforms standard holomorphic equilibrium propagation.

**Strengths:**

The method improves over previous holomorphic equilibrium propagation in the applications tested.

As someone who is not in this field, I found section 2 to be informative introduction to the area.

**Weaknesses:**

It was not entirely clear to me what the motivation for this work is.  While the authors suggest that hEP is potentially biologically plausible, this seems like a stretch.  Not only does it require neurons to do computations with complex numbers, it also seems to require that the network settle to equilibrium at multiple phases of an ongoing oscillation.  This does not seem likely in the brain: the period of the gamma oscillation is tens of milliseconds, which certainly would not allow enough time, and even the theta oscillation seems to fast for this. It also was not clear what computational advantages this might lead to in purely artificial systems for which biological plausibility was not important.

**Questions:**

Most important for me is to explain how this could be helpful in artificial learning systems, or why the problem of settling time does not apply to biological systems (which seems unlikely to be honest).

---

> ### Author Response · Authors · 2023-11-14
> **Individual response to reviewer 2L2m**
>
> > *“As someone who is not in this field, I found section 2 to be informative introduction to the area.”*
>
> We thank reviewer 2L2m for the positive comment.
> ___
> > *“While the authors suggest that hEP is potentially biologically plausible, this seems like a stretch. [...] it also seems to require that the network settle to equilibrium at multiple phases of an ongoing oscillation”*
>
> We agree that for now the question of bio-plausibility is debatable. But here are our arguments.
>
> On a conceptual level, the brain does use a hierarchy of oscillations spanning a whole range of frequency, as reviewer 2L2m pointed out, so the fact that holomorphic EP introduces oscillation in a principled way to do learning is a very promising similarity in our opinion. Yet, answering the question of plausibility for hEP will require investigating potential *mappings* from the complex units to neurons or microcircuits in the brain. For instance, ReLU activation is often interpreted as a firing rate, and a mapping to spiking network can be made based on that. For complex-valued activation, one can imagine that it encodes not only a firing rate, but also a phase with respect to an underlying rhythm. Another promising advantage that makes hEP plausible shown in the original paper (Laborieux & Zenke 2022) is the robustness to noise in the dynamics, which is not only required for any theory of learning in the brain, but also useful for neuromorphics artificial learning systems.
>
> About the question on whether the settling time is an issue for biological systems, we want to mention that neurons could use a mechanism called “prospective dynamics”, which means they try to anticipate their future activity, and it allows them to tightly track a moving equilibrium, such that the lag is reduced (see [r1], [r2] and references therein), this mechanism is compatible with our theory where neurons could track the oscillating teaching signal and reduce the time needed to learn.
> ___
> > *“Most important for me is to explain how this could be helpful in artificial learning systems”*
>
> We list below some of the reasons why EP, and in particular the holomorphic extension could be helpful in artificial learning systems such as neuromorphic hardware [r3]:
> 1. The system only needs to handle one type of computation: the neural dynamics, and it need not handle a separate linear computation to derive gradients (as in BP). Designing a system that has to handle both computation with the same memory devices is more difficult than just one computation.
> 2. The learning rule in hEP only involves the activity of pre and post neurons, and therefore is local, which is a decisive advantage for building an efficient artificial learning system.
> 3. The robustness to noise already mentioned in the previous point. Not only is it bio-plausible, but also useful for noisy neuromorphic hardware.
> 4. Another advantage is that the error is encoded as an integral of significantly varying (non-vanishing) neuronal oscillations, rather than a finite difference between two nearby states, which is difficult to sense in noisy substrates. In short, the gradient in hEP is computed accurately despite the network oscillating in a nonlinear way, while other approaches typically rely on a linear expansion around the free fixed point.
> 5. In addition, the error as an integral can be accumulated in continuous time, removing the need to access the gradient at a precise oscillation period.
> 6. Last but not least, robustness to asymmetric weights in the extent we describe in the paper.
>
> Finally, hardware realization leveraging EP suggests that it can dramatically reduce the energy cost of AI training by several orders of magnitude [r3]. Taken together, we hope that these reasons will convince reviewer 2L2m that our work is also useful for artificial systems.
> ___
> **References**
>
> [r1] Haider, Paul, et al. "Latent equilibrium: A unified learning theory for arbitrarily fast computation with arbitrarily slow neurons." Advances in Neural Information Processing Systems 34 (2021): 17839-17851.
>
> [r2] Senn, Walter, et al. "A neuronal least-action principle for real-time learning in cortical circuits." bioRxiv (2023): 2023-03.
>
> [r3] Yi, Su-in, et al. "Activity-difference training of deep neural networks using memristor crossbars." Nature Electronics 6.1 (2023): 45-51.

---

> > ### Comment · Reviewer_2L2m · 2023-11-22
> >
> > Thanks for the response.  I remain unconvinced about bioplausibility.  Adding the comments on artificial systems would improve the manuscript, but not enough to warrant a change in the review scores.

---

> > > ### Author Response · Authors · 2023-11-22
> > >
> > > Points well-taken, we will tone down the claims on bioplausibility in favor of highlighting the benefits for artificial systems. Thanks for your time.

---

### Official Review · Reviewer_nv6j · 2023-10-31

**Soundness:** 3 good
**Presentation:** 3 good
**Contribution:** 2 fair
**Rating:** 6
**Confidence:** 3

**Summary:**

This paper studied the weight asymmetry in the weight transport problem extending the Holomorphic Equilibrium Propagation algorithm (hEP), through investigations about the Jacobian of the network, with experiments and theoretically analysed the reasons for the experimental results. Based on the analysis results, some new features were proposed to enable the algorithm to evade the need for perfect weight symmetry without affecting the functional symmetry. A new form of Jacobian homeostasis was introduced to maintain functional symmetry, without directly addressing weight symmetry. Finally, several experiments, including the investigation of weight symmetry evolution during training and comparative experiments, were conducted to verify the effectiveness of the proposed algorithm, and the work’s performance exceeded the networks with Recurrent Backpropagation (RBP), even on larger datasets.

**Strengths:**

*Relatively new perspective

This paper shows a new perspective on studying the weight transport problem with the Jacobian of the network. It explains the connections between weight symmetry, functional symmetry and Jacobian homeostasis.

*Relatively rigorous analysis and persuasive experiments

This paper shows the efforts in investigating the weight symmetry evolution during training in section 4. If the observations could be discussed more deeply with the figure would be better.

*The part about ‘Jacobian homeostasis improves functional symmetry’ in section 4 is very detailed and well analysed.

**Weaknesses:**

*Some expressions with flaws

Although ‘hEP’ is the abbreviation of ‘holomorphic Equilibrium propagation’ can be understood after reading. However, this abbreviation has not been indicated in parenthesis when mentioning its full term for the first time.
The last sentence ‘It is worth nothing that…’, in Definition 1 of section 2.2, might be miswritten, which should be ‘It is worth noting that…’ based on the context. Also in this paragraph, the sentence after Eq. (6) ‘Importantly, the quantities …, which applies only to EBMs.’, is a little longer and complex. It might confuse readers, for convenience of understanding, it would be better to split it into two simpler sentences.

*Some disadvantages in the layout

In section 3.2, the paragraph with Eq. (9) is intersected by Figure 2, which could be rearranged to provide enhanced readability. The same disadvantages happen in Figure 3 and Table 1.

*The performance of the proposed method is relatively not so advanced. And the method for comparison is relatively not so new, making the work of this article not very convincing.

**Questions:**

Please refer to the above weakness.

---

> ### Author Response · Authors · 2023-11-14
> **Individual response to reviewer nv6j**
>
> We thank reviewer nv6j for the positive feedback and comments.
> ___
> > *“This paper shows the efforts in investigating the weight symmetry evolution during training in section 4. If the observations could be discussed more deeply with the figure would be better.”*
>
> We understand that reviewer nv6j means that we should discuss more in detail Fig. 4 c,g,k. For Fig 4 c,g, we record a metric quantifying the degree of symmetry of the Jacobian $J$. By expressing the Jacobian as a sum of a symmetric matrix and an antisymmetric matrix $J = S + A$, we record the quantity $\frac{\| S \|}{\|S\| + \|A \|}$ which varies from 0 to 1 where 0 means fully antisymmetric (provided $J \neq 0$), and 1 means fully symmetric. In Fig 4c we see that the metric goes very close to one because the connections are reciprocal between layers (Fig 4a). For Fig 4g, the metric increases but stays far from 1 because the connections are not reciprocal (Fig 4e), so perfect symmetry is unachievable. For Fig 4k, because we use a larger network (Fig 4i), computing the symmetry metric as in Fig. 4 c,g is not possible (the jacobian is too large a matrix), so instead we record the angle between forward and feedback weights as a replacement, for which 0 now means perfect symmetry. We see that the homeostatic loss makes the angle smaller and therefore improves symmetry, while still being significantly different from perfect symmetry. If reviewer nv6j is happy with these explanations, we will update Section 4 accordingly.
> ___
> > *“Although ‘hEP’ is the abbreviation of ‘holomorphic Equilibrium propagation’ can be understood after reading. However, this abbreviation has not been indicated in parenthesis when mentioning its full term for the first time.”*
>
> Thanks for catching this important detail, it has now been fixed.
> ___
> > *“The last sentence ‘It is worth nothing that…’, in Definition 1 of section 2.2, might be miswritten”*
>
> Thanks for reading carefully, the typo has been fixed.
> ___
> > _“the sentence after Eq. (6) ‘Importantly, the quantities …, which applies only to EBMs.’, is a little longer and complex. It might confuse readers, for convenience of understanding, it would be better to split it into two simpler sentences.”_
>
> We acknowledge that the sentence was a bit heavy, so we have separated it into two simpler sentences.
> ___
> > *“the paragraph with Eq. (9) is intersected by Figure 2, which could be rearranged to provide enhanced readability. The same disadvantages happen in Figure 3 and Table 1.”*
>
> We thank reviewer nv6j for pointing these out, we have changed the position of the Figure 2 one page earlier so that it does not intersect the paragraph with Eq. (9). We also move Figure 3 and Table 1 so that it does not intersect the paragraph.
> ___
> > *“The performance of the proposed method is relatively not so advanced.”*
>
> We agree that our method is tailored to EP and dynamical networks, for which the performance is behind state-of-the-art approaches that use other architectures. We reworked our introduction (in blue in the updated pdf) to justify our approach of focusing on EP primarily because it is a promising algorithm for efficient learning hardware. Despite lower performance at the moment, we believe that the results we present are necessary to further improve the performance of EP approaches in the future.

---

### Official Review · Reviewer_8uWs · 2023-10-31

**Soundness:** 4 excellent
**Presentation:** 2 fair
**Contribution:** 2 fair
**Rating:** 8
**Confidence:** 4

**Summary:**

_Disclaimer: I have reviewed this paper recently again at a different venue. It has since been revised slightly. Some of my comments still apply to this version and are copied here verbatim from my previous review._

The paper focuses on Equilbrium Propagation (EP), i.e. a learning algorithm for neural networks that was introduced relatively recently and aspires to be more biologically plausible and more suitable for analog hardware than back-propagation (BP). The theoretical version of EP relies on an infinitesimal perturbation (which is not feasible to achieve in physical settings) and on symmetric weights (which are also unrealistic in biology and constrain hardware design). The paper aims to understand the impact of each of these two issues separately. It achieves to do so in the "holomorphic EP" (hEP) setting, i.e. under the assumption of complex-valued neurons. The authors achieve this by extending the theory of hEP to asymmetric connectivity, where they show that it is possible to obtain exact estimates of the gradient despite finite perturbations, therefore suggesting that the bigger issue is weight asymmetry. They then proceed to tackle the problem of asymmetry by introducing a loss term that penalizes it directly, and they show experimentally that this term improves the performance of hEP significantly in asymmetrically-initialized networks, in certain visual tasks using a 4-layer convolutional network.

**Strengths:**

The paper is mostly nicely written and clear. The theoretical contributions of the paper are not trivial, as they require a deep understanding of a very specific algorithm i.e. the holomorphic version of EP, as well as a degree of comfort with certain mathematical concepts that is rare among neural network practitioners and possibly even theoreticians. More generally, the paper aims to contribute to an area that is of broad interest, as it relates to machine learning, neuromorphic engineering, and theoretical neuroscience. Furthermore, it truly advances the empirical results of the EP-related literature.

**Weaknesses:**

While this is obviously a valuable piece of work in certain respects, I believe it is also significantly limited in other key aspects.

Significance: The work ultimately aspires to improve aspects of backpropagation that are indeed important and relevant to multiple large disciplines (ML, neuromorphic hardware, neuroscience), however, concretely, the resulting contribution is very narrow. Namely, it improves the performance and the theoretical understanding of hEP somewhat, specifically in the case of asymmetric weights, but it doesn't resolve the problem of asymmetry completely, as can be seen in the performance comparison to symmetric weights (Table 2). Moreover, hEP is a specialized version of EP that makes additional assumptions for complex-valued networks, which limits the applicability and generality of the algorithm. Furthermore, EP itself more broadly is interesting but is a rather limited method in terms of achieving its goals of good performance, efficiency, useful hardware demonstrations, or deep learning, in comparison with other methods that have similar goals that have been more successful. Its biological compatibility could also be debated, given its requirement for multiple network-wide iterations before a weight update. Therefore, it could be argued that the significance of the results only relates to a very narrow subfield. Furthermore, the theoretical advance separating the impact of the finite nudge from that of the asymmetry in hEP is not trivial, but I wonder how useful it is to the ICLR community, beyond the very narrow sub-community that specializes in hEP. In the broader picture, and given the already existing better-working alternatives, the progress made here towards biological plausibility or neuromorphic computation seems small, in my view.

Novelty: The main novelty of the paper is in the theoretical results, since the empirical fact that improving performance by dealing with asymmetric weights is not a new result. Nevertheless, I suspect that to someone with a good understanding of the earlier mathematical work that introduced hEP (Laborieux & Zenke, NeurIPS 2022) the new theoretical results might not be very surprising. However, it should be noted that I am not in a position to judge this fully. In any case, the insight from the theory is the impact of asymmetrical weights, relative to that of finite nudge, which seems to be also covered by the empirical result, therefore the added value from the theory in this case might not be substantial. Regarding the paper's empirical or practical contributions, some novelty exists in the objective function that penalizes asymmetries in the weight matrix. However, the novelty of this is limited because much older learning rules that achieve weight symmetry do exist (Kolen & Polak, IJCNN 1994; see also Payeur et al., Nat. Neurosci. 2021). In fact, these rules do not rely on global iterative equilibria, so the present paper's implementation of learnable weight symmetry could be characterized as a step-back in this regard. It should be recognize though that the new method is applicable when the connectivity is not reciprocal, whereas previous ones probably were not.

Contextualization in the literature: The paper does (now) cite some of the works that had similar aims, but only in passing, only in the discussion as opposed to the motivation section, discounts the better empirical results that the other works achieved, attributing this only to the lower simulation cost of the alternatives, and does not mention that some of these methods not only perform better in classification benchmarks, but also require fewer assumptions for compatibility with biology and for efficiency in learning hardware, e.g. by circumventing the need for backward passes of information completely.
Some examples are: Payeur et al., Nat. Neurosci. 2021; Greedy et al., NeurIPS 2022; Mengye Ren et al., ICLR 2023; Journé et al., ICLR 2023.

It would be very helpful to the readers and the targeted research communities if the authors motivated their choice to focus specifically on EP as opposed to alternative methods that have similar goals, but don't have the same limitations. This could be a way to mitigate the weaknesses of the paper to some extent.

To be clear, some of the algorithm's limitations in comparison to alternatives are: reliance on complex-valued networks; constraints in network architecture and depth, expensive simulations for the equilibrium dynamics (e.g. a multi-GPU cluster was used according to the supplementary material, despite the small networks and simple tasks); questionable bio-plausibility of the necessity for equilibrium; hardware implementations of EP are mostly theoretical. To their credit, the authors have mentioned some of these limitations in the discussion. They also provided some solutions or counterarguments, however these are largely theoretical, vague, or speculative.

All in all, I believe that the work's value might be able to increase if the manuscript could explain its motivations and its contributions in a context broader than the EP literature. At present, this is not clear enough or supported well enough, in my view.

**Questions:**

The discussion suggests that "analog substrates could achieve this relaxation “for free” through device physics (Yi et al., 2023; Kendall et al., 2020)". Could the authors please clarify, does any analog substrate achieve the relaxation "for free", or what are the requirements, more specifically? For example, the Kendall et al. reference seems to rely on a very peculiar and ad hoc hardware implementation.

Also, in the most efficient hypothetical hardware implementation, I suspect that "for free" is far from the truth, as, even there, the relaxation phase would include multiple weight updates, which consume power; in fact commonly more power than weight reads. Wouldn't alternative algorithms that do not have this relaxation phase be significantly more efficient? The provided references do not seem to disagree with this, so in which sense do the references support the "free" claim?

I would appreciate the authors' insight in these questions.

---

> ### Author Response · Authors · 2023-11-14
> **Individual response to reviewer 8uWs (1/2)**
>
> We thank reviewer 8uWs for having enough interest in our work to review it for the second time. We are also thankful for the positive comments about the writing, the high mathematical standards, as well as the value of our work. Below we reply point by point to the points raised by reviewer 8uWs:
> ___
> > “_[...] the resulting contribution is very narrow. [...] hEP is a specialized version of EP that makes additional assumptions for complex-valued networks [...]  the results only relates to a very narrow subfield [...] I wonder how useful it is to the ICLR community, beyond the very narrow sub-community that specializes in hEP_”
>
> We still respectfully disagree, and believe that our results are of interest for all the broader community studying alternative learning algorithms. For instance, we see at least three subfields different from EP, and represented at ICLR and similar conferences, for which our results are interesting:
> 1. Predictive coding and predictive coding networks, which are another kind of EBM often trained with algorithms related to EP (e.g. Millidge et al ICLR 2023)
> 2. Deep equilibrium models, which also rely on iterative equilibria for different purposes as EP, but with transferable insights (e.g. Bai, Koltun & Kolter ICLR 2021)
> 3. Target propagation, which is interested in mechanisms for improving symmetry (Meuleumans et al NeurIPS 2020, 2021, 2022)
> ___
> > *“It would be very helpful to the readers and the targeted research communities if the authors motivated their choice to focus specifically on EP as opposed to alternative methods that have similar goals, but don't have the same limitations. [...] I believe that the work's value might be able to increase if the manuscript could explain its motivations and its contributions in a context broader than the EP literature.”*
>
> We focus on EP for two main reasons:
>
> 1. Because of its potential to reduce the energy cost of AI training when running on a specific hardware substrate (such as already demonstrated in Yi et al 2023), which is a broader subject than the EP literature.
> 2. Because EP is also a promising model for modeling learning in the brain (local learning rule, oscillations), though we do concede that a concrete mapping is still missing to make this statement compelling, which we acknowledge in the discussion.
>
> For this reason, we choose to focus on EP and reduce the impact of asymmetry and finite nudge on its performance, rather than systematically comparing it to other methods. We believe it is fair since we do demonstrate non trivial performance compared to BP for similar architectures, and on challenging tasks such as ImageNet 32.
>
> **we have updated the introduction to highlight that EP running on dedicated hardware, as demonstrated in Yi et al 2023, can reduce the training cost of AI training by four orders of magnitude**. We believe that this statement places our work in the context of reducing AI’s energy footprint, which is broader than EP itself, and also constitutes the main reason why we choose to focus on EP.
> ___
> > *“The paper does (now) cite some of the works that had similar aims, but only in passing, only in the discussion as opposed to the motivation section, discounts the better empirical results that the other works achieved, attributing this only to the lower simulation cost of the alternatives, and does not mention that some of these methods not only perform better in classification benchmarks, but also require fewer assumptions for compatibility with biology and for efficiency in learning hardware, e.g. by circumventing the need for backward passes of information completely.”*
>
> Apology, by no means did we mean to discount other methods with our explanation for EP’s lower performance, and are sorry if it was interpreted this way. Our reasoning was that since EP has a theoretical link with gradients computed by BP, it makes little doubt that higher performances with EP can be achieved with better engineering, which is corroborated by this very recent paper [r1] that demonstrated better runtime and performance for EP. **We have now removed this argument, and updated the introduction to mention explicitly that other bio plausible alternatives well-suited for learning hardware, do perform better than EP**.
>
> [r1] Scellier et al *“Energy-based learning algorithms for analog computing: a comparative study”* NeurIPS 2023

---

> > ### Comment · Reviewer_8uWs · 2023-11-19
> >
> > > our results are of interest for all the broader community studying alternative learning algorithms
> >
> > My understanding is that the results don't even hold for EP itself, but rather only for the complex-number-reliant hEP algorithm. Is that so? In that case, how do the results help with even more distant algorithms? Can this contribution be concretized? Do this manuscript's theoretical results help in any way there? Has the empirical method for mitigating weight asymmetry been validated with any other algorithm than hEP?
> >
> >
> > > We focus on EP for two main reasons: [...] 1. reduce the energy cost of AI training [...] 2. for modeling learning in the brain
> >
> > That is very clear. What is not clear is why these reasons led the authors to hEP and not to any of the other algorithms that are equally well - if not better - suited for aims 1. and 2. I am not necessarily suggesting that this cannot be justified. However, I am suggesting that this is not made clear at all in the paper, even though proper contextualization and motivation would demand it.

---

> > > ### Author Response · Authors · 2023-11-20
> > >
> > > > *My understanding is that the results don't even hold for EP itself, but rather only for the complex-number-reliant hEP algorithm. Is that so? In that case, how do the results help with even more distant algorithms?*
> > >
> > > The main difference between multiple variants of EP (classic one-sided [r1], centered [r2] etc) and hEP is only in how the neuronal error vector $\delta$ is computed/estimated, not in the symmetry/asymmetry of the weights. Therefore, **our results about mitigating weight asymmetry do apply to all these variants equally**. More broadly, they apply to more distant algorithms as they also fall under the same convergent dynamical system framework we describe in Section 2.1 (predictive coding networks, deep equilibrium models), so the notion of network Jacobian we use is exactly the same in those works, and our method can be applied as is.
> > >
> > > > *Can this contribution be concretized? Do this manuscript's theoretical results help in any way there? Has the empirical method for mitigating weight asymmetry been validated with any other algorithm than hEP?*
> > >
> > > It definitely can be concretized. In the time we have left, we will work on an additional experiment to demonstrate the empirical method for mitigating weight asymmetry on classic EP and predictive coding networks.
> > > ___
> > > > *What is not clear is why these reasons led the authors to hEP and not to any of the other algorithms that are equally well - if not better - suited for aims 1. and 2.*
> > >
> > > One contribution of our work is to derive the hEP gradient estimate when the weights are asymmetric (Eqs 6,7,8,9), which was not done in the original hEP paper where the estimate was only derived in terms of the energy function, enforcing weight symmetry. This contribution requires focusing on hEP rather than other algorithms. More generally, we think hEP is promising enough regarding our aims to focus on it despite the existence of concurrent algorithms, that we fully acknowledge in introduction now, as requested previously by reviewer 8uWs.
> > >
> > > ___
> > > **References**
> > > [r1] Scellier, Benjamin, and Yoshua Bengio. "Equilibrium propagation: Bridging the gap between energy-based models and backpropagation." Frontiers in computational neuroscience 11 (2017): 24.
> > > [r2] Laborieux, Axel, et al. "Scaling equilibrium propagation to deep convnets by drastically reducing its gradient estimator bias." Frontiers in neuroscience 15 (2021): 633674.

---

> ### Author Response · Authors · 2023-11-14
> **Individual response to reviewer 8uWs (2/2)**
>
> > *“[...] it doesn't resolve the problem of asymmetry completely [...] the empirical fact that improving performance by dealing with asymmetric weights is not a new result. [...] the new theoretical results might not be very surprising [...] It should be recognized though that the new method is applicable when the connectivity is not reciprocal, whereas previous ones probably were not.”*
>
> We stress again that our claim is to mitigate the effect of asymmetry in the context of EP, rather than solving it completely. We still do believe that our way of improving symmetry for EP is novel and surprising, since it does not simply amount to making forward and backward weights identical, like in earlier solutions such as Kolen & Polack. We thank reviewer 8uWs for acknowledging that. Finally, because our new method is general, it can transfer to other subfields represented at ICLR (enumerated above).
> ___
> > *“Could the authors please clarify, does any analog substrate achieve the relaxation "for free", or what are the requirements, more specifically? For example, the Kendall et al. reference seems to rely on a very peculiar and ad hoc hardware implementation. [...] even there, the relaxation phase would include multiple weight updates”*
>
> Sorry for the confusion. By “for free”, we meant that the laws of physics bring the substrate to equilibrium, such that the computation need not be implemented explicitly as in traditional von Neumann hardware. In Kendall et al, this is done with a nonlinear resistive network as a physical substrate, and the corresponding laws of physics responsible for doing the computation are the Kirchhoff’s laws. In particular, no weight update is required for the relaxation phase (see paragraph “Performing inference” in Kendall et al). The fact that their hardware implementation is very peculiar and hoc is on purpose. The idea is to co-design a hardware dedicated to a specific training algorithm (EP), and not a general purpose hardware. We regret the poor choice of words “for free” and **have removed it from the discussion.**
> ___
> > *“Wouldn't alternative algorithms that do not have this relaxation phase be significantly more efficient?”*
>
> Good question. The question of efficiency is not only a question of algorithm, but also of hardware substrate on which it runs. We are not saying that EP will be more efficient than other algorithms on conventional computers, we are saying that EP is well suited to be implemented on a dedicated hardware substrate, such as Yi et al 2023, because computation can be carried out efficiently with physical laws, unlike alternative methods, to the best of our knowledge. Owing to these advantages, it can be more efficient than other algorithms running on conventional computers (cf Yi et al 2023), but we agree that this prospect is somewhat remote for now. Nonetheless, it is still a research direction worth pursuing and being represented at ICLR.
> ___
> > *“To be clear, some of the algorithm's limitations in comparison to alternatives are: [...] multi-GPU cluster was used according to the supplementary material, despite the small networks and simple tasks [...] To their credit, the authors have mentioned some of these limitations in the discussion. They also provided some solutions or counterarguments, however these are largely theoretical, vague, or speculative.”*
>
> We are sorry that reviewer 8uWs is not convinced by our arguments. However, we do mention most of these limitations in the discussion:
> - Complex numbers: “oscillate in the complex plane, whose physical interpretation is not straightforward in neurobiology”
> - Iterative equilibria: “Another limitation of EP is that it requires convergence to an equilibrium.”
> - Costly simulations: “This requirement makes EP costly on digital computers.”
>
> As for the network depth, and unless we are mistaken, it is also not significantly shallower than Journé et al and Payeur et al who test on five or so layers CNNs.
>
> About task simplicity, we want to stress again that we do test on ImageNet 32, which does not qualify as a “simple task”, and is not significantly less complex than standard ImageNet, since only the resolution changes, not the number of datapoints or classes.

---

> > ### Comment · Reviewer_8uWs · 2023-11-19
> >
> > > Good question
> >
> > I would like to clarify that my question was indeed about dedicated hardware, where computation is carried out by physical laws, i.e. it can be complemente as follows: “Wouldn't alternative algorithms that do not have this relaxation phase be significantly more efficient [even on dedicated hardware]?”

---

> ### Author Response · Authors · 2023-11-20
>
> > *Wouldn't alternative algorithms that do not have this relaxation phase be significantly more efficient [even on dedicated hardware]?*
>
> It is difficult to give a definite answer without further details on the specific algorithm and hardware.
>
> For the sake of example, let's consider a crossbar hardware architecture which can do matrix multiplication with Kirchhoff's law, and compare both BP and EP.
>
> Implementing a traditional ANN with BP does not require a relaxation phase, but requires a record of the operations done in the forward pass to compute the backward pass, moreover, the backward pass in BP transmits the $\delta$ directly, which is easily disrupted by the noise of the hardware due to its scale (see paragraph "backpropagation or errors" in [r1]).
>
> On the other hand, EP performs implicit differentiation through the nudge, the details of the operation during the relaxation phase are not needed, because $\delta$ would not be transmitted explicitly as in BP, but implicitly through the change of neural activity, which is agnostic to the hardware imperfections since the same dynamics is used.
>
> So in this case, even though an algorithm such as BP has no relaxation phase, the details of the hardware and the requirement of the algorithm itself make it less suited than EP despite the relaxation phase [r2].
>
> We hope this example can convince reviewer 8uWs about the validity of researching EP-related algorithms.
> ___
> **References**
> [r1] Narayanan, Pritish, et al. "Toward on-chip acceleration of the backpropagation algorithm using nonvolatile memory." IBM Journal of Research and Development 61.4/5 (2017): 11-1.
> [r2] Yi, Su-in, et al. "Activity-difference training of deep neural networks using memristor crossbars." Nature Electronics 6.1 (2023): 45-51.

---

> ### Comment · Reviewer_8uWs · 2023-11-21
>
> In my perception, these responses evade my actual questions.
>
> - The authors still have not explained how the *theoretical* results can be extended to anything other than hEP, even EP.
> If that is not possible, then my judgment remains that the theoretical contribution is very narrow. For completeness, I think that the *practical* result of partly dealing with asymmetry is somewhat useful but not particularly surprising, or especially impactful in light of prior methods that deal with asymmetry. The promised added experiments would add some value.
>
> - The authors do not seem to be able to motivate researchers of biological learning or of neuromorphic hardware to study EP as opposed to other alternatives to backpropagation.
>
> The above are my main criticisms of the paper.
>
> The below is merely a question seeking clarification, but the authors also evaded it.
>
> - The last reply by the authors compared with BP even though the question was about hardware-oriented *alternatives* to BP that do not have backward passes of any sort and do not have multiple iterations per training example. Again, it appears to me that such alternatives would be more efficient than EP, which requires multiple forward and backward passes and multiple weight update iterations for each training example.

---

> > ### Author Response · Authors · 2023-11-21
> >
> > > *The authors still have not explained how the theoretical results can be extended to anything other than hEP, even EP*
> >
> > The homeostatic loss we present is part of the theoretical results (Section 3 of the paper), and we explained that it can be used we classic EP, predictive coding networks, and deep equilibrium models, because they all fall under the framework of converging dynamical systems. Therefore, our theoretical contribution is not narrow and not limited to hEP.
> >
> > To be clear, our theoretical contributions (section 3) are:
> > - Expression of the EP error vector as an integral (Section 3.1)
> > - Estimation of the error vector and pre-synaptic terms as an integral spanning multiple periods (Section 3.2)
> > - Bias due to asymmetry (Section 3.3), which also applies to classic EP.
> > - Homeostatic loss (section 3.4) which applies to EP, hEP, predictive coding networks, deep equilibrium models.
> >
> > > *The authors do not seem to be able to motivate researchers of biological learning or of neuromorphic hardware to study EP as opposed to alternatives to backpropagation.*
> >
> > Even though EP and hEP have practical drawbacks of iterative equilibria and complex numbers for hEP, that we already acknowledge, they also do have compelling features regarding biological learning and neuromorphic hardware that motivate  research on EP. We list them below for clarity:
> >
> > 1. The same network dynamics handle the computation of a theoretically sound loss gradient
> > 2. Local learning rule involving only pre and post neuron activity
> > 3. Robustness to noisy dynamics demonstrated in the original hEP paper.
> > 4. Error encoded as an integral rather than a finite difference.
> > 5. The error as an integral can be accumulated over multiple periods to get rid of precise timing.
> > 6. There already exists prototypes to run EP-like learning algorithms (Yi et al 2023) to cut the cost of AI training by orders of magnitude
> >
> > We hope reviewer 8uWs can understand that these points motivate research on EP, despite the existence of other concurrent algorithms, which we acknowledge in the introduction following previous comments.
> >
> > > *The last reply by the authors compared with BP even though the question was about hardware-oriented alternatives to BP*
> >
> > There are _many_ BP alternatives but only one BP, which is why we made this comparison for sake of argument. We are happy to discuss more in details if reviewer 8uWs points out a specific alternative to BP.
> >
> > We also provided multiple times a reference to Yi et al 2023 which actually builds hardware learning with a method based on EP, with demonstrated energy savings compared to GPUs. We are still wondering what is reviewer 8uWs' take on this. For instance, do the other BP alternatives reviewer 8uWs is thinking about have been demonstrated in such hardware or to such extent?
> >
> > For reference, here is a quote of Yi et al abstract:
> >
> > > Our approach, which we term memristor activity-difference energy minimization, treats the network parameters as a constrained optimization problem, and numerically calculates local gradients via Hopfield-like energy minimization using behavioural differences in the hardware targeted by the training. We use the technique to train one-layer and multilayer neural networks that can classify Braille words with high accuracy. With modelling, we show that our approach can offer over four orders of magnitude energy advantage compared with digital approaches for scaled-up problem sizes.

---

> ### Comment · Reviewer_8uWs · 2023-11-21
>
> - The clarification that Sections 3.1, 3.2, 3.3 could also apply to classic EP, not only hEP, is very helpful.
> It is not clear in the paper that this is the case, or how this is the case, as these sections seem to only mention hEP.
> Can this be mitigated?
>
> - In a further attempt to help the authors motivate the manuscript, is any of those listed nice features of EP unique to EP? I do not think so. There are certainly unique disadvantages, as it has been agreed on in this discussion. Again, are there any unique advantages to EP (already listed or not)? If that can be specified, it would be very helpful.
>
> - Efficient Hebbian learning on chip has been demonstrated to a much larger extent than EP, in multiple occasions, and in multiplle renditions, for decades now. Here is an example from 1992: https://ieeexplore.ieee.org/document/173117
> Here is another example in a spiking implementation that also has a power estimate (4mW) https://www.frontiersin.org/articles/10.3389/fnins.2015.00141/full
> Here is an example with a memristive implementation https://www.nature.com/articles/s41467-018-04933-y
> Other local learning algorithms too. Here is a chip for e-prop consuming 50 µW https://ieeexplore.ieee.org/document/9731734
> These are only examples, there are possibly hundreds of such publications.
> Again, it appears that the authors have focused on EP without acknowledging its context.

---

> > ### Author Response · Authors · 2023-11-21
> >
> > > *The clarification that Sections 3.1, 3.2, 3.3 could also apply to classic EP, not only hEP, is very helpful. It is not clear in the paper that this is the case, or how this is the case, as these sections seem to only mention hEP. Can this be mitigated?*
> >
> > Thanks for the positive feedback. **We will mitigate this by mentioning in section 3 that the homeostatic loss can be applied to the other approaches we mentioned.** We hope that it will make it more clear to the reader that our results do not apply only to hEP.
> >
> > > *is any of those listed nice features of EP unique to EP?*
> >
> > While some of these features taken individually are maybe not unique to EP, we believe that the accumulation of all those features into a single algorithm is unique, to the best of our knowledge.
> >
> > In particular, the Cauchy integral formulation of the error vector we present offers the potential to compute the error through _nonlinear_ oscillations. We are not aware of any other algorithm that can compute a theoretically sound error (that is, linked to an _actual_ loss gradient), by integrating _nonlinear_ dynamics. Typically, many other approaches like classic EP, predictive coding networks, target prop use _finite differences_ to compute the error, which means they need that the network varies only linearly with respect to its prediction. In hEP, the oscillations can vary significantly around the free fixed point, such that the nonlinearity of the network is harnessed rather than fought against, which is an important feature for dedicated hardware.
> >
> > > *Efficient Hebbian learning on chip has been demonstrated to a much larger extent than EP*
> >
> > Thanks for providing references, some of which we were not aware of. **To improve contextualization of EP, we will mention those hardware work and some others in the introduction.**
> >
> > We note however that it is unclear whether the algorithms implemented on those chips would outperform EP or not, since they are sometimes bottom-up STDP-based and/or not tested on the same tasks as ours (for instance ImageNet 32). We understand that the work mentioned before as those outperforming EP (Payeur et al., Greedy et al., Mengye Ren et al. ; Journé et al.) have not been implemented on dedicated hardware yet, unlike EP (Yi et al). We therefore think that EP remains an algorithm worth researching.

---

> ### Comment · Reviewer_8uWs · 2023-11-21
>
> > In particular, the Cauchy integral formulation of the error vector we present offers the potential to compute the error through nonlinear oscillations. We are not aware of any other algorithm that can compute a theoretically sound error (that is, linked to an actual loss gradient), by integrating nonlinear dynamics. Typically, many other approaches like classic EP, predictive coding networks, target prop use finite differences to compute the error, which means they need that the network varies only linearly with respect to its prediction. In hEP, the oscillations can vary significantly around the free fixed point, such that the nonlinearity of the network is harnessed rather than fought against, which is an important feature for dedicated hardware.
>
> This is indeed an important advantage of hEP over other approaches that also rely on oscillations.
>
> I am now convinced that the paper is a useful contribution to the literature of oscillation-based algorithms, which is itself important.
> I strongly recommend that the paper's introduction and conclusions make clear that the motivation, the focus and the contributions are in this subfield, which is narrower than the broader field of biologically plausible learning theory and of neuromorphic learning algorithms. I have not been convinced that the broader disciplines have advanced significantly in this paper.
>
> Could this improvement be made?
> If so, then along with the other improvements that the authors have committed to, I would raise my score.
> Could the authors please briefly recap the changes they will make?
>
>
>
> Secondarily, and for the completeness of our discussion:
> > While some of these features taken individually are maybe not unique to EP, we believe that the accumulation of all those features into a single algorithm is unique, to the best of our knowledge.
>
> That is only true if the comparison is limited among oscillation-based networks. Again, other algorithms that do not have this requirement for multiple iterations have all these advantages and more as has been mentioned a few times during this discussion.
>
> > We understand that the work mentioned before as those outperforming EP (Payeur et al., Greedy et al., Mengye Ren et al. ; Journé et al.) have not been implemented on dedicated hardware yet, unlike EP (Yi et al).
>
> Yi et al's implementation is memristor activity-difference energy minimization (MADEM) which is very closely related to EP, but not exactly the same. Similarly, among the multiple existing realizations of physical on-chip learning, some are very closely related to, for example, the Hebbian winner-take-all of Journé et al. that the authors just mentioned. E.g. Kreiser et al.'s https://ieeexplore.ieee.org/document/8325168 and https://www.nature.com/articles/s41467-018-04933-y.

---

> > ### Author Response · Authors · 2023-11-22
> >
> > > *This is indeed an important advantage of hEP over other approaches that also rely on oscillations.
> > I am now convinced that the paper is a useful contribution to the literature of oscillation-based algorithms, which is itself important.*
> >
> > Thanks for the positive feedback, we agree to edit the intro and conclusions to frame our results specific to hEP as contributions to oscillation based learning algorithms.
> >
> > Here is a final recap of the proposed changes:
> >
> > - Frame hEP results as contribution to oscillation based algorithms.
> > - Better contextualization of EP hardware through citation of other hardware approaches.
> > - Moderate the claims about bio-plausibility.
> > - Highlight that the theory of homeostatic loss also applies to classic EP, predictive coding nets, and deep equilibrium models.  We will add an appendix section demonstrating the benefits of the homeostatic loss for PCNs.
> >
> >  And also the ones already mentioned in the global response:
> >
> > - To better explain the motivation of our work, we edited the introduction to explain that we focus on EP not for peak performance, but because of its strong potential to reduce the energy cost of AI training, as e.g. demonstrated by Yi et al 2023.
> > - To better place our work with respect to other approaches, we now acknowledge in the introduction that other bio-plausible algorithms report better peak performance than EP.
> > - Pending: a better explanation of Fig 4 regarding weight symmetry evolution during training.
> > - We changed the layouts so that Figures and Table don’t overlap with text, for better readability.
> > - We removed confusing terminology from the discussion.
> > - We properly introduce the hEP acronym.
> > - We fixed the various typos.
> >
> > We believe that these changes will substantially improve the quality of our paper, and thank reviewer 8uWs for the discussion.

---

> > > ### Comment · Reviewer_8uWs · 2023-11-22
> > > **Raising my score**
> > >
> > > With these changes, assuming they are implemented properly, this will be one of the few papers in this subfield that argue entirely rigorously and honestly among many that often make very weakly supported strong claims about biology and AI hardware. This on its own makes the paper valuable.
> > >
> > > Moreover, the paper does offer interesting theoretical insight and useful practical results to the community of oscillation-based algorithms.
> > >
> > > Based on these, I have raised my score.

---

### Official Review · Reviewer_EQJi · 2023-11-05

**Soundness:** 3 good
**Presentation:** 3 good
**Contribution:** 3 good
**Rating:** 6
**Confidence:** 4

**Summary:**

Equilibrium propagation, an alternative to backprop requires weight symmetry and nudges to yield unbiased gradient estimates. Generalization of Equilibrium proportion to non-symmetric dynamical system exists but shown to work only for simple problems.
The paper proposes an extension of holomorphic EP to non-symmetrical dynamical systems and shows good results across different vision benchmarks.

**Strengths:**

- Analysis of bias in gradient estimation is helpful to the reader.
- Incorporating functional symmetry through the use of matching jacobians is an interesting idea. It’s similar to reconstruction error term used in methods like Target Propagation. Here, authors optimize the homeostatic loss with respect to all the weights as compared to using only feedback weights.

**Weaknesses:**

- The paper is generally well written, though introduction is a bit complex if the reader is not aware of previous work (holomorphic EP).

**Questions:**

No further questions as such. The paper has good results across different vision benchmarks.

---

> ### Author Response · Authors · 2023-11-14
> **Individual response to reviewer EQJi**
>
> > *“introduction is a bit complex if the reader is not aware of previous work (holomorphic EP).”*
>
> Thanks for the feedback, we rewrote the sentence introducing holomorphic EP in the introduction to make it easier for the reader. It is highlighted in blue in the revised version.
>
> > *“No further questions as such. The paper has good results across different vision benchmarks.”*
>
> We thank reviewer EQJi for the positive comment.

---

### Author Response · Authors · 2023-11-14
**Global response**

We thank the reviewers for the time spent in reviewing our work, as well as the positive comments and constructive feedback. We summarize below a list of changes to the paper. We already highlighted some of the changes in blue in the updated pdf.

___

**Enhancements and modifications to the revised paper:**

1. To better explain the motivation of our work, we edited the introduction to explain that we focus on EP not for peak performance, but because of its strong potential to reduce the energy cost of AI training, as e.g. demonstrated by Yi et al 2023.
2. To better situate our work with respect to other approaches, we now acknowledge in the introduction that other bio-plausible algorithms report better peak performance than EP.
3. Pending: a better explanation of Fig 4 regarding weight symmetry evolution during training.
4. We changed the layouts so that Figures and Table don’t overlap with text, for better readability.
5. We removed confusing terminology from the discussion.
6. We properly introduce the hEP acronym.
7. We fixed the various typos.

We provide below more detailed answers to each reviewer. We remain available to answer any further questions, and are looking forward to constructive discussions.

---

### Meta-Review · Area_Chair_mfcw · 2023-12-10

**Metareview:**

Equilibrium propagation is a learning algorithm that has advantages over backpropagation (e.g. locality, hardware implementation), but it requires weight symmetry to function properly. The paper introduces a new method to reduce the negative effects of asymmetric weights via a homeostatic normalization of the jacobian, and demonstrates that it substantially improves learning performance.

The reviews exhibit some spread in their scoring (5,6,6,8) but all state that the proposed method is clever and sophisticated, the implementation is technically sound, and that the work is clearly presented and well written. Negatives mostly concerned the claimed biophysical plausibility and the perceived rather narrow focus. The authors, however, managed to convincingly explain the more broader implications of their work and also its connections to the relevant literature, which was incorporated in the revision of the paper.

The ACs overall recommendation takes into account that the reviewer who gave the highest score also provided by far the most detailed and informative assessment. This reviewer also showed great understanding of the work, and maintained a constructive and interesting dialogue with the authors. Also, as a result of this discussion the reviewer raised his initial score. On the other hand, the reviewer with the lowest score is clearly not an expert on the matter, and the criticism was limited to concerns about the biophysical plausibility of the learning algorithm, which is fair but does not deter from the interest of the work to the ML community.

**Justification For Why Not Higher Score:**

The paper should be accepted. However, a higher score is probably not warranted given that the proposed method reduces the bias induced by asymmetric weights, but ultimately does not solve the problem. Furthermore, although the implications of using a homeostatic normalization constraint are likely to have implications for other local learning algorithms broader, the authors did not show those in the work but considered the more narrow case of holomorphic equilibrium propagation.

**Justification For Why Not Lower Score:**

The paper addresses an important limitation of class of local learning algorithms. It is technically sophisticated and sound. It is of general interest for the ML community. It is well written.

---

### Decision · Program_Chairs · 2024-01-16

Accept (poster)